# Precise capture and dynamic relocation of nanoparticulate biomolecules through dielectrophoretic enhancement by vertical nanogap architectures

Eui-Sang Yu [1,2], Hyojin Lee[3], Sun-Mi Lee[4], Jiwon Kim[4], Taehyun Kim [1], Jongsu Lee[1], Chulki Kim[1], Minah Seo [1], Jae Hun Kim[1], Young Tae Byun[1], Seung-Chul Park[5], Seung-Yeol Lee[6], Sin-Doo Lee [2✉] & Yong-Sang Ryu [1✉]

Toward the development of surface-sensitive analytical techniques for biosensors and diagnostic biochip assays, a local integration of low-concentration target materials into the sensing region of interest is essential to improve the sensitivity and reliability of the devices. As a result, the dynamic process of sorting and accurate positioning the nanoparticulate biomolecules within pre-defined micro/nanostructures is critical, however, it remains a huge hurdle for the realization of practical surface-sensitive biosensors and biochips. A scalable, massive, and non-destructive trapping methodology based on dielectrophoretic forces is highly demanded for assembling nanoparticles and biosensing tools. Herein, we propose a vertical nanogap architecture with an electrode-insulator-electrode stack structure, facilitating the generation of strong dielectrophoretic forces at low voltages, to precisely capture and spatiotemporally manipulate nanoparticles and molecular assemblies, including lipid vesicles and amyloid-beta protofibrils/oligomers. Our vertical nanogap platform, allowing low-voltage nanoparticle captures on optical metasurface designs, provides new opportunities for constructing advanced surface-sensitive optoelectronic sensors.

[1] Sensor System Research Center, Korea Institute of Science and Technology, Seoul 02792, Republic of Korea. [2] Department of Electrical and Computer Engineering, Seoul National University, Seoul 08826, Republic of Korea. [3] Center for Biomaterials, Korea Institute of Science and Technology, Seoul 02792, Republic of Korea. [4] Clean Energy Research Center, Korea Institute of Science and Technology, Seoul 02792, Republic of Korea. [5] Department of Nature-Inspired Nanoconvergence Systems, Korea Institute of Machinery and Materials, Daejeon 34103, Republic of Korea. [6] School of Electronics Engineering, Kyungpook National University, Daegu 41566, Republic of Korea. ✉email: sidlee@plaza.snu.ac.kr; ysryu82@kist.re.kr

Bridging the gap between bulky materials and atomic/molecular structures sub-100-nm nanoparticles (NPs) are of great scientific interest owing to their extraordinary physical properties, for instance, significant enhancement of reactivity[1], collective oscillation of electrons on localized surfaces[2], and quantum confinement[3]. In particular, owing to the potential role of nanoparticulate biomaterials in unveiling sub-cellular biological mechanisms[4], the capturing and trapping techniques of the bio-NPs in specific positions have been intensively studied not only to identify biological phenomena[5,6] but also for next generation applications as drug delivery carriers[7], imaging agents[8], bioelectronic devices[9], analytic lab-on-a-chip systems[10], and biosensors[11]. Despite their increasing demand, manipulation of NPs is extremely challenging owing to the enhanced disruptive Brownian noise with reduction in NP sizes[12]. A variety of techniques have been developed to manipulate the spatiotemporal positions of NPs using external stimuli, including optical[13,14], fluidic[15-17], mechanical[18,19], magnetic[20,21], and electrical biases[22-25]. Among them, the usefulness of electrical techniques has been amply proven by easy actuation as well as uniform stimulation over large areas in a label-free manner. However, for conventional architectures with microgap electrodes, high voltage application is indispensable for trapping the suspended NPs in an aqueous environment, which inevitably accompanies the undesirable formation of electrolytic bubbles and the denaturation of biomaterials by Joule heating[26].

As a promising alternative for microgap architecture, nanogap electrodes were previously utilized for NP capturing in low-voltage schemes[27-29]. Although NPs were well captured, sub-10-nm nanogap electrodes suffer from electron tunneling through a thin insulator[30]. Thus, a new strategy other than the reduction of insulator thickness is required to enhance the capacity for NP trapping. Conventionally, nanogap electrodes were prepared using expensive and sophisticated manufacturing methods that limited their shape and area. Small-area applicability and low-degrees-of-freedom in electrode designs have prohibited high-throughput molecular detection with nanogap structures as well as possible application on surface-sensitive optoelectronic sensors with various spectral ranges and enhanced sensitivities. Therefore, a platform that enables the implantation of optical metasurfaces along with dynamic manipulations of the bio-NPs on their sensing areas will offer opportunities to develop advanced surface-sensitive biosensors that are yet to be realized.

Herein, we report unique and versatile vertical nanogap electrode (VNE) that enables capturing and relocating bio-NPs, with large-area applicability and high degrees-of-freedom in electrode design flexibility (Fig. 1a). The VNE array, which is designed to generate well-controlled flow dynamics accompanied with improved particle capture, consists of a patterned insulator layer sandwiched between two electrode layers (one plane and one patterned; Fig. 1b). Enhanced performance was experimentally demonstrated by capturing sub-100-nm NPs in a sub-volt application to confirm NP capturing in a size-selective manner and relocate the bio-NPs, including nanometer-scale lipid vesicles and peptide assemblies, along with a theoretical analysis of electrokinetic behavior.

## Results

### Forces acting on suspended particles and resultant particle dynamics.

Suspended particles under an AC electric field (E-field) experience dielectrophoresis (DEP)[31], AC electro-osmosis (ACEO)[32], and other various external forces. The ACEO flow is caused by the movement of surface-attracted ions, which results from a thin layer of induced charge on the solid–liquid interface called the electrical double layer (EDL). Although the ACEO can

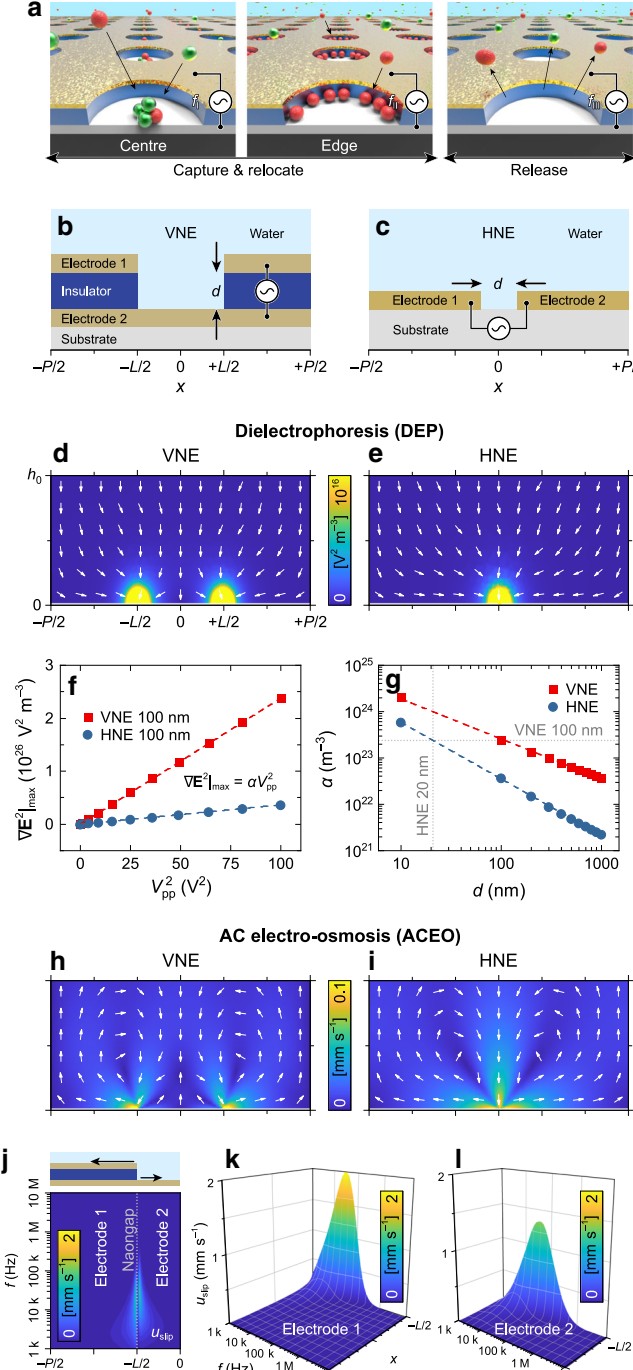

**Fig. 1 Dielectrophoretic and AC electro-osmotic characterization on VNE.**
**a** Conceptual illustration of capture and relocation of NPs toward the center (left) and edge of VNE (middle) and their release (right). **b, c** Schematic of the VNE (**b**) and HNE (**c**). **d, e** Calculated distribution of $\nabla E^2$ on VNE (**d**) and HNE (**e**) with a gap distance of $d = 100$ nm under $V_{pp} = 1$ V ($h_0 = 10$ μm). **f, g** Calculated $\nabla \mathbf{E}^2|_{max}$ on VNE (red square) and HNE (blue circle) as a function of $V^2_{pp}$ (**f**), and its corresponding $\alpha$ as a function of $d$ (**g**). **h, i** Calculated distribution of ACEO flow on VNE (**h**) and HNE (**i**) with a gap distance of $d = 100$ nm under $V_{pp} = 1$ V and $f = 1$ kHz. **j–l** Calculated $u_{slip}$ distribution at the electrode surfaces of VNE ($d = 100$ nm) as a function of $f$ (**j**, $V_{pp} = 1$ V) with the schematic of $u_{slip}$ (inset in **j**) and 3D illustrations of $u_{slip}$ on electrode 1 (**k**, top electrode) and 2 (**l**, bottom electrode).

be applied to sort, transport, and position the suspended particles[33], it can barely manipulate the particles in a material- or size-selective manner. Unlike the ACEO, the DEP force can selectively trap and expel suspended particles from the electrodes where the maximum E-field gradient is generated[34]. For a spherical dielectric particle of radius $R$ suspended in a medium of permittivity $\varepsilon_m$, the time-averaged $\mathbf{F}_{DEP}$ under an AC voltage frequency of $f$ can be expressed as[35]

$$\mathbf{F}_{DEP} = \pi\varepsilon_m R^3 \mathrm{Re}[f_{CM}(\omega)] \cdot \nabla\mathbf{E}_{rms}^2, \qquad (1)$$

where $\mathbf{E}_{rms}$ and $\omega$ are the root mean-squared E-field and angular frequency ($\omega = 2\pi f$), respectively. The term $\mathrm{Re}[f_{CM}(\omega)]$ denotes the real part of the Clausius–Mossotti (CM) factor, which represents the effective polarizability of the particle in a medium. The CM factor is given by

$$f_{CM}(\omega) = \frac{\varepsilon_p{}^*(\omega) - \varepsilon_m{}^*(\omega)}{\varepsilon_p{}^*(\omega) + 2\varepsilon_m{}^*(\omega)}, \qquad (2)$$

in which $\varepsilon_p{}^*$ and $\varepsilon_m{}^*$ are the complex permittivities of the particle and medium, respectively, and are expressed by their respective permittivities ($\varepsilon_p$, $\varepsilon_m$) and conductivities ($\sigma_p$, $\sigma_m$); $\varepsilon^* = \varepsilon - i \cdot (\sigma/\omega)$, where $i = \sqrt{-1}$. According to Eqs. (1) and (2), controlling the frequency and voltage amplitude serve to selectively capture the NPs according to material and size, respectively. To evaluate the competitive forces and resultant dynamics of a spherical particle (with its mass of $m_p$) in an aqueous environment, the Langevin equation of particle velocity ($\mathbf{u}_p$) was employed[35]:

$$m_p \frac{d\mathbf{u}_p}{dt} = \mathbf{F}_{DEP} + \mathbf{F}_{ACEO} + \mathbf{F}_{ETF} + \mathbf{F}_{grav} + \mathbf{F}_{buoy} + \mathbf{F}_{int} + \xi(t),$$

$$(3)$$

considering DEP force ($\mathbf{F}_{DEP}$), Stokes drag force induced by ACEO flow ($\mathbf{F}_{ACEO}$) and electrothermal flow ($\mathbf{F}_{ETF}$), gravitational force ($\mathbf{F}_{grav}$), buoyant force ($\mathbf{F}_{buoy}$), interparticle force from Coulomb interaction ($\mathbf{F}_{int}$), and random Brownian force ($\xi(t)$). Because Stokes drag forces are defined as $\mathbf{F} = -6\pi\eta R(\mathbf{u}_p - \mathbf{u}_m)$, where $\mathbf{u}_p$ and $\mathbf{u}_m$ are the velocities of particles and fluidic flows, respectively, $\mathbf{F}_{ACEO}$ and $\mathbf{F}_{ETF}$ exerting on a single particle can be evaluated by inserting $\mathbf{u}_{ACEO}$ and $\mathbf{u}_{ETF}$ into $\mathbf{u}_m$, respectively[36]. Considering the simulations ($\mathbf{F}_{ETF}$, $\mathbf{F}_{DEP}$, and $\mathbf{F}_{ACEO}$), calculations ($\mathbf{F}_{grav}$, $\mathbf{F}_{buoy}$, and $\xi(t)$), and low-concentration (10 ppm) condition ($\mathbf{F}_{int}$) on the VNE having pattern size ($L$) and periodicity ($P$) of $L = 10\ \mu m$ and $P = 30\ \mu m$ (Fig. 1b), $\mathbf{F}_{DEP}$ and $\mathbf{F}_{ACEO}$ dominantly determine the movement of 1-μm-diameter polystyrene (PS) particles, while the others are negligible (Supplementary Notes 1–5, Supplementary Table 1, and Supplementary Figs. 1–4). As the relaxation time ($\tau$) of the particle ($\tau = m/6\pi\eta R \approx 6 \times 10^{-8}$ s) is much smaller than the typical experimental observation time ($t$), transient ballistic regime of short time scales ($t \ll \tau$) quickly reaches viscous regime of a time scale longer than $\tau$ ($t \gg \tau$). Thus, Eq. (3) can be approximated into terminal $\mathbf{u}_p$ of viscous regime as ref. [36]

$$\mathbf{u}_p = \left(\mathbf{u}_{ACEO} + \frac{\mathbf{F}_{DEP}}{6\pi\eta R}\right)\left(1 - e^{-t/\tau}\right) \approx \mathbf{u}_{ACEO} + \frac{\mathbf{F}_{DEP}}{6\pi\eta R} \qquad (4)$$

with the assumption that the initial velocity of the particle is 0. From a dimensionless analysis of particles that exhibit small Stokes numbers (Supplementary Note 6), the viscous fluidic effect is expected to dominate over the negligible inertial effect and the motions of particles will be coupled with flow. Therefore, the movements and positions of the suspended particles under various AC conditions can be anticipated by calculating two dominant electrokinetics of DEP and ACEO.

## Dielectrophoretic- and AC electro-osmotic characterization and fabrication of VNE

According to Eqs. (1) and (2), $\mathbf{F}_{DEP}$ is determined by physical factors such as particle size, $\pi R^3$, frequency-dependent electrical properties of the particles and media, $\varepsilon_m \mathrm{Re}[f_{CM}(\omega)]$, and a gradient of the E-field squared, i.e., $\nabla\mathbf{E}^2$. Unlike other operating factors, $\nabla\mathbf{E}^2$ is determined by the structural characteristics of the electrodes. For the capture of smaller particles, the generation of higher values of $\nabla\mathbf{E}^2$ is advantageous for dielectrophoretic performance in particle capture. From this perspective, we evaluated $\nabla\mathbf{E}^2$ of VNE under various peak-to-peak voltages ($V_{pp}$) and compared the dielectrophoretic performance to $\nabla\mathbf{E}^2$ of conventional horizontal nanogap electrode (HNE) (Fig. 1c), with an identical gap distance of $d = 100$ nm using COMSOL Multiphysics software. According to the simulation results, $\nabla\mathbf{E}^2$ is found to be strongest at the nanogap and weakens farther away from the edges (Fig. 1d, e). By extracting and plotting the maximum value of $\nabla\mathbf{E}^2$ ($\nabla\mathbf{E}^2|_{max}$) as a function of $V^2_{pp}$ (Fig. 1f), the linear relation between these two parameters can be expressed as $\nabla\mathbf{E}^2|_{max} = \alpha V^2_{pp}$, where the slope of the plot ($\alpha$) quantitatively represents the dielectrophoretic performance. Based on this analysis, it was found that VNE generated stronger $\nabla\mathbf{E}^2$ with more dramatic proportionality to the voltage than the HNE. For example, in the case of $d = 100$ nm, the simulated value of $\alpha$ from the VNE ($\alpha_{VNE}$) was $\alpha_{VNE} = 2.4 \times 10^{23}$ V$^2$ m$^{-3}$, whereas that from the HNE ($\alpha_{HNE}$) was $\alpha_{HNE} = 3.7 \times 10^{22}$ V$^2$ m$^{-3}$. This indicates that VNE generates 6.5 times stronger $\mathbf{F}_{DEP}$ under identical voltage application and that HNE requires a 2.6 times larger voltage amplitude to generate identical trapping performance. This is attributed to the asymmetric structure of the VNE electrodes, i.e., the bottom electrode is a plane whereas the top one is patterned, inducing higher non-uniformity of the E-field than symmetrically arranged HNEs. Furthermore, to test whether $\mathbf{F}_{DEP}$ of VNEs are enhanced at various gap distances, we conducted similar processes for $d = 10$ to 1000 nm (Fig. 1g). The calculated results clearly show that values of $\alpha_{VNE}$ are 3.4 ($d = 10$ nm) to 16.5 ($d = 1000$ nm) times higher than $\alpha_{HNE}$. This demonstrates the improved DEP performance of VNE in particle capturing irrespective of the gap distance, although its enhancement decreases as $d$ becomes narrower. From this simulation result, it is presumed that the VNE with $d = 100$ nm achieves an equivalent DEP performance to that of HNE with $d = 20$ nm. After establishing the superiority of VNEs in terms of DEP performance, the thermal Joule heating effect was investigated to confirm the viability of a biomaterial in the presence of E-fields. According to the simulated results for both VNE and HNE under various $V_{pp}$, a negligible temperature increment ($\Delta T$) of <0.04 K was expected in the presence of $V_{pp} = 10$ V (Supplementary Note 7 and Supplementary Fig. 5).

Together with DEP performance, ACEO-driven local slip velocities at the boundary of two electrodes and resultant bulk fluidic flow across the insulator of both VNE and HNE were explored. The simulations of hydrodynamic bulky flow over the each electrode revealed that VNE generates a pair of microvortices rotating in opposite directions with distinct magnitudes (Fig. 1h), in contrast to symmetric HNEs, which yield bulky flows of equivalent magnitude in an outward direction from the nanogap (Fig. 1i). As is well known, the slip velocity, $u_{slip}$, arising at the electrode surface from the interaction of the E-field and EDL, drives the bulky fluidic motions. We analyzed $u_{slip}$ to characterize ACEO behavior as a function of both frequency and position on the VNE surface (Fig. 1j–l). On the boundary of the VNE surface, $u_{slip}$ reached a maximum at the edge and became decreased away from the nanogap irrespective of the applied frequency (Fig. 1j). Although the simulation results show good correspondence with those of previous studies examining ACEO-assisted slip velocity above horizontally aligned electrodes[32,37–39],

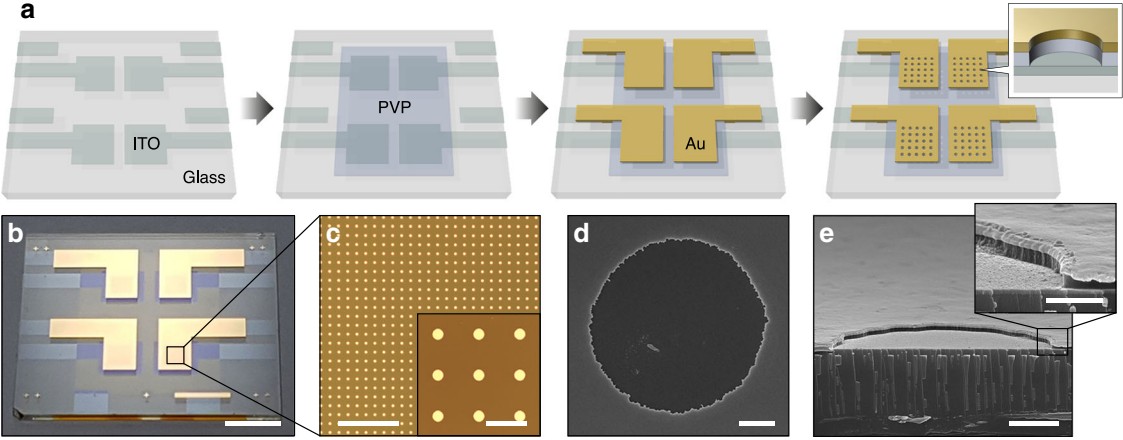

**Fig. 2 Fabrication of VNE arrays. a** Schematics of the fabrication process. **b–e** A VNE sample (**b**), micrographs of a VNE array (**c**), top-view (**d**), and cross-sectional view SEM images (**e**) of a VNE single unit. Scale bars, 5 mm (**b**), 200 μm (**c**), 30 μm (inset in **c**), 2 μm (**d**, **e**), and 500 nm (inset in **e**).

the VNE offers unique hydrodynamic flows resulting in asymmetry in magnitudes of $u_{slip}$ at each micro-vortex (Fig. 1k, l). This imbalance of a faster $u_{slip}$ on the top electrode than that on bottom electrode arises from the structural configuration of VNE, as previously demonstrated by the use of asymmetric horizontal electrode arrays[40–42] or non-planar ones[43–45]. Owing to this faster $u_{slip}$ on the top electrode, the micro-vortex on the top electrode induces larger flow streamlines than those on the bottom electrode (Fig. 1h). Moreover, this imbalance is intensified when the disparity of each flow across the vertical nanogap increases under different frequency conditions, and this hints at the frequency-dependent distinct dynamics of particles in combination with DEP (Supplementary Note 8, Supplementary Figs. 6 and 7).

Based on the theoretical results, VNE arrays with $d = 100$ nm were fabricated over a large area to demonstrate their structural fidelity and excellent production yields (Fig. 2, Supplementary Note 9, and Supplementary Fig. 8). On a glass substrate, a 40-nm-thick indium tin oxide bottom electrode was patterned and spin-coated with a 100-nm-thick insulating layer of poly(4-vinyl phenol) (PVP) and thermal crosslinking agent. Subsequently, a 40-nm-thick Au electrode was thermally deposited through a shadow mask. Next, the Au and insulating layers were patterned into 10-μm-hole arrays via photolithography and sequential etching processes (Fig. 2a). The photograph (Fig. 2b), micrographs (Fig. 2c), and scanning electron microscope (SEM) images (Fig. 2d, e) clearly demonstrate that uniform device architectures of well-controlled 100-nm-thick nanogap electrodes were achieved over a large area.

**Simulated particle dynamics under AC frequency.** After designing the VNE arrays, we evaluated how AC frequencies affect the movement of suspended particles through two dominant forces of $\mathbf{F}_{ACEO}$ and $\mathbf{F}_{DEP}$ ($\mathbf{F}_{total} = \mathbf{F}_{ACEO} + \mathbf{F}_{DEP}$, Fig. 3). We classify the frequency ranges into 3 regimes: low-frequency (1–10 kHz; regime I), mid-frequency (0.1–1 MHz; regime II), and high frequency (> 10 MHz; regime III), where each regime is expected to be dominantly influenced by ACEO, positive DEP (pDEP), and negative DEP (nDEP), respectively. The frequency-dependent dynamics of suspended particles on a single unit of VNE were simulated using a combination of ACEO and DEP (Fig. 3a–e) and experimentally monitored for cross-validation (Fig. 3f–j). Note that the electrohydrodynamic drag forces acting on the particle by ACEO were calculated using $\mathbf{F}_{ACEO} = -6\pi\eta R (\mathbf{u}_p - \mathbf{u}_{ACEO})$, with assumptions that E-field had just been applied and that the particle was at rest prior to the AC

application ($\mathbf{u}_p = 0$)[46]. The DEP-induced velocity of particles ($\mathbf{u}_{DEP}$) was also calculated and simultaneously expressed in the form of $\mathbf{u}_{DEP} = \mathbf{F}_{DEP}/6\pi\eta R$ for direct comparison with $\mathbf{u}_{ACEO}$. To observe the real-time particle movement through bright-field (BF) microscopy, we used 1-μm-diameter PS particles with a concentration of $1.8 \times 10^8$ particles ml$^{-1}$ and simulated the phenomena using physical values of $V_{pp} = 2.5$ V, $\varepsilon_m = 80 \cdot \varepsilon_0$, $\sigma_m = 1$ μS cm$^{-1}$, $\varepsilon_p = 2.55 \cdot \varepsilon_0$, $\sigma_p = 114$ μS cm$^{-1}$, and $R = 500$ nm. In regime I, pairs of asymmetric ACEO flows circulating above each surface of VNE were simulated both in 1 and 10 kHz. Because ACEO is predominant in the low-frequency range of regime I, the movements of particles are dominantly determined by ACEO rather than DEP. Within regime I, distinct particle dynamics are also expected with respect to the applied frequencies owing to the varying dominant forces; ACEO is dominant over DEP at 1 kHz (Fig. 3a), whereas they are comparable at 10 kHz (Fig. 3b), as shown through simulations (Supplementary Fig. 4). In contrast, the formation of the EDL layer is weakened at high frequency as the alternation of electrical signals is too rapid for ions to follow; thus, particles are majorly governed by $\mathbf{F}_{DEP}$ (Fig. 3c–e).

The direction of $\mathbf{F}_{DEP}$ can also be controlled by changing the frequency because $\mathbf{F}_{DEP}$ highly depends on the polarizing behavior of particles. For example, particles move along the increasing field gradient at the condition of pDEP where Re$[f_{CM}(\omega)] > 0$, whereas repulsion of particles from the region of the highest field gradient is referred to as nDEP where Re$[f_{CM}(\omega)] < 0$[34]. In particular, for charged particles with sizes comparable to their EDL thickness, ionic currents and their convection within the EDL play an important role in polarization and formation of induced dipole[47]. To evaluate the dielectric response of a suspended particle in the presence of an applied E-field, the Maxwell–Wagner–O'Konski (MWO) model was adopted to consider both local surface charges and the resultant surrounding EDL. In this model of MWO dielectric dispersion, the effective conductivity of particles ($\sigma_p$) with a radius of $R$ can be described as $\sigma_p = \sigma_{bulk} + 2K_{surf}/R$, which is a combination of the bulk conductivity ($\sigma_{bulk}$) and surface conductance ($K_{surf}$) of particles[48]. This implies that NPs with high surface area to volume ratio experience higher $K_{surf}$; thus, surface conductance become dominant in the case of smaller NPs. Because MWO theory works at low conductive environments[49,50], the effective dielectrophoretic properties of PS NPs, including CM factor, were represented using the MWO model (Supplementary Note 10, Supplementary Table 2, and Supplementary Fig. 9). The crossover from positive to negative values of Re$[f_{CM}(\omega)]$ occurs at a specific crossover frequency ($f_\chi$), which is 1.8 MHz in this case. Because

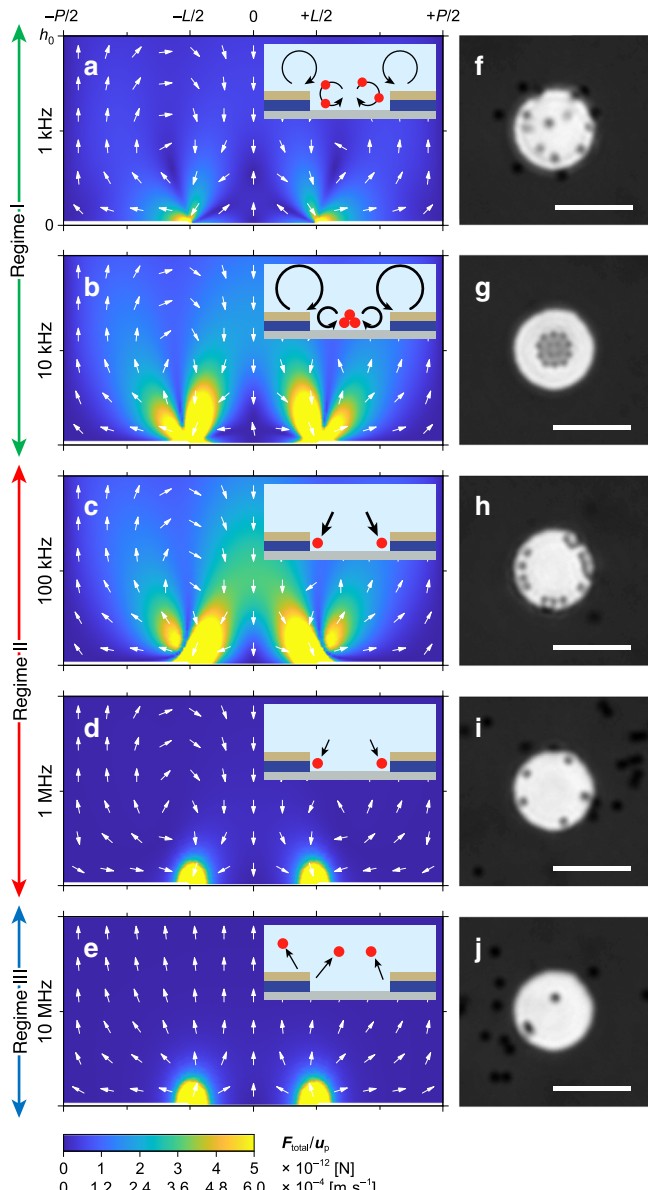

**Fig. 3 Frequency-dependent particle movements on the VNE. a–j** Cross-sectional simulations of $\mathbf{F}_{total}$ and $\mathbf{u}_p$ with schematics (**a–e**) and top-view of the experimental BF micrographs (**f–j**) of suspended 1 μm PS particles on a single VNE under various AC frequencies; $V_{pp} = 2.5$ V, $f = 1$ kHz (**a, f**), 10 kHz (**b, g**), 100 kHz (**c, h**), 1 MHz (**d, i**), and 10 MHz (**e, j**). Scale bars, 10 μm.

Re[$f_{CM}(\omega)$] value decreases as the frequency approaches $f_\chi$, the pDEP force at 100 kHz (Fig. 3c) is found to be stronger than that at 1 MHz (Fig. 3d). Above 1.8 MHz, the directions of $\mathbf{F}_{DEP}$ become reversed; thus, repulsion of particles from the nanogap is expected in regime III at 10 MHz (Fig. 3e).

**Experiments of particle capture and relocation.** Based on the simulation analysis of the particle trajectories on VNEs, corresponding experiments were conducted using AC voltages of $V_{pp} = 2.5$ V (Fig. 3f–j and Methods). At 1 kHz (Fig. 3f), the particles were swirling over the single unit of VNE because they are significantly influenced by ACEO flows rather than the considerably weaker DEP force. However, at 10 kHz, suspended particles are migrated to the center of the unit cell, where flows are converged to form a point of stagnation (Fig. 3g and

Supplementary Fig. 4). Unlike at 1 kHz, an application at 10 kHz generates both ACEO and DEP with comparable magnitudes. With an aid of attracting $\mathbf{F}_{DEP}$, particles are brought into proximity with the VNE electrode surface against upward ACEO flow. Then, because the ACEO-induced slip velocity (pushing toward center) is found to be stronger than the DEP force (attracting toward edges) at the collecting surface of the VNE, effective surface streams push particles toward the stagnation point where the pseudo-potential energy is the lowest (Supplementary Note 11 and Supplementary Fig. 10)[51]. In contrast, at 100 kHz (Fig. 3h) and 1 MHz (Fig. 3i) in regime II, the ring-shaped particle assembly was observed at the edge of nanogap, where the value of $\nabla\mathbf{E}^2$ is the largest and the pseudo-potential energy is the lowest (Supplementary Note 11 and Supplementary Fig. 10). Although ACEO flows still exist at 100 kHz, trapping at the stagnation deteriorates as a pair of asymmetric micro-vortices vanish into a single large flow and $\mathbf{F}_{DEP}$ becomes stronger than $\mathbf{F}_{ACEO}$ (Supplementary Fig. 4). Instead, a long-range ACEO flow constantly conveys distant particles to the VNE, and the dominant $\mathbf{F}_{DEP}$ successfully snatches the moving NPs from the flow and accumulate them to the edge of VNEs at 100 kHz. Note that the interparticle force component ($\mathbf{F}_{int}$) is negligible when suspended particles are distant from each other. However, when they are trapped and come together (Fig. 3g, h), particle–particle interactions improve the trapping efficiency of the dielectrophoresis devices by the induced dipole moments of particles[35,52,53]. As predicted above, a small amount of particles trapped (Fig. 3i) at 1 MHz indicates a weaker $\mathbf{F}_{DEP}$ (Fig. 3d) generated by a smaller Re [$f_{CM}(\omega)$] value compared to that at 100 kHz (Fig. 3c). Lastly, the particles are dislodged from the edge of the pattern by the nDEP force at 10 MHz in regime III (Fig. 3j). Although the particles can be collected at much lower voltage amplitudes, a significantly strong voltage of 2.5 V was applied to clarify the recognizable particle relocation by the frequency changes (Supplementary Movie 1). Furthermore, to investigate the minimum voltage and effective trapping volume, corresponding simulations and experiments were conducted by varying the voltage amplitudes at a fixed frequency of 100 kHz. When $V_{pp} = 0.1$ V, particles started to get captured and gradually populated the nanogap regions, and more particles were collected by increasing the voltage amplitude (Supplementary Note 12, Supplementary Figs. 11 and 12). Note that 0.1 V may not be the minimum trapping threshold, as our waveform generator was not able to supply a lower voltage. As an example of micron-scale bioparticles, the possible capturing ranges of yeast cells and *Bacillus subtilis* spores were theoretically evaluated using spherical- and elongated multi-shell dielectric models using MWO theory, and the trapping behaviors were experimentally confirmed under low-voltage amplitudes of $V_{pp} = 0.5$ and 0.8 V, respectively (Supplementary Notes 13 and 14, Supplementary Tables 3 and 4, and Supplementary Figs. 13–16).

**Size-selective dielectrophoretic capture of NPs.** After establishing the general principles of particle movement under the influence of two dominant electrokinetic forces in each frequency regime, the size-selective trapping and relocation of submicron particles in the vicinity of the nanogap were investigated further in the VNE (Fig. 4). Because $\mathbf{F}_{DEP}$ is proportional to the volume of particle, $\propto \pi R^3$, (Eq. (1)), particles of different sizes could be selectively captured when they reached their respective threshold voltages for trapping. To examine the relationship between NP's size and minimum voltages, we prepared a mixture solution of PS particles with diameters of 200, 100, and 50 nm tagged in different fluorescent (FL) dyes of blue (DAPI), green (FITC), and red (Texas Red), respectively. According to the calculation of the CM factor based on the MWO model, the values of $f_\chi$ for each NP

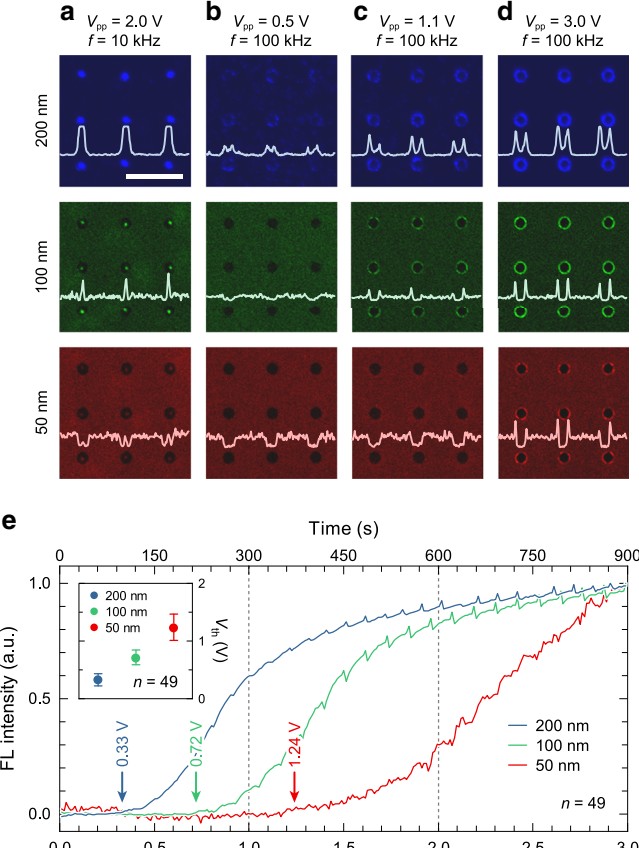

**Fig. 4 Size-selective trapping of PS NPs at various AC voltages. a–d**
Experimental micrographs of captured PS NPs with diameters of 200 nm
(first row; blue), 100 nm (second row; green), and 50 nm (third row; red)
under $V_{pp} = 2$ V at $f = 10$ kHz (**a**), $V_{pp} = 0.5$ V (**b**), 1.1 V (**c**), and 3 V (**d**) at
$f = 100$ kHz. Scale bar, 50 μm. **e** Averaged FL intensities of captured PS NPs
on 49 unit cells as a result of increasing $V_{pp}$ ($f = 100$ kHz) from 0 to 3 V by
0.1 V for every 30 s; averaged trapping thresholds of 0.33, 0.72, and 1.24 V
for 200, 100, and 50 nm, respectively (inset).

were calculated as 12, 18, and 37 MHz, respectively (Supplementary Note 10 and Supplementary Fig. 9). From this analysis, a frequency of 10 kHz was selected to achieve ACEO dominance, and 100 kHz guaranteed DEP dominance. For the real-time monitoring of NP capture and relocation, confocal FL microscopy was used to compare size-by-size capture for different AC voltage amplitudes (Fig. 4). At $V_{pp} = 2$ V and $f = 10$ kHz, all NPs were collected at the center of the circular pattern owing to the combination of $\mathbf{F}_{ACEO}$ and $\mathbf{F}_{DEP}$, and at the same time, the FL intensity, which indicates the population of NPs, were proportional to their size (Fig. 4a). Because the magnitude of $\mathbf{F}_{DEP}$ is greater for larger particles, 200 nm particles tend to be more populated than smaller ones. This explains the role of $\mathbf{F}_{DEP}$ on size-selective concentration at 10 kHz. In contrast, at 100 kHz, only 200 nm PS particles started to be captured at the nanogap from the averaged trapping threshold ($V_{th}$) of $V_{th} = 0.33$ V (Fig. 4b, e). As the AC voltage increased, the 100 nm particles began to be captured at $V_{th} = 0.72$ V while no 50 nm particles were monitored (Fig. 4c, e). Finally, the trapping of the 50 nm particles was observed for $V_{th} = 1.24$ V (Fig. 4d, e) while other larger particles became highly packed (Supplementary Note 15 and Supplementary Fig. 17). In this process, the time-lapse normalized FL intensities were analyzed and averaged for a VNE unit cell number ($n$) of $n = 49$, and each standard deviation (s.d.) of

$V_{th}$ was calculated to be 0.11, 0.12, and 0.22 V for PS NPs of 200, 100, and 50 nm, respectively (indicated as error bars in inset of Fig. 4e). Note that a glitch occurred at the moment when electrical signals changed owing to equipment limitations. The results clearly confirm that the VNE can also spatially relocate the NPs in a size-selective manner by controlling the frequency and magnitude of AC bias. That is, the frequency determines the local position of NPs whereas the control of voltage magnitude enables their selective capture according to size.

**Low-voltage trapping of nano-vesicles in pre-defined VNE geometry.** Beyond the artificially synthetized NPs, the trapping behavior of nanoparticulate biomaterial was investigated using 50-nm-diameter small unilamellar vesicles (SUVs). SUVs, which are spherical vesicles bounded by a lipid bilayer, are widely used to study biological phenomena by mimicking extracellular vesicles (EVs), which are important mediators of intercellular communications in biological activities[54]. The SUVs were composed of 1,2-Dioleoyl-sn-glycero-3-phosphatidylcholine (DOPC) and 3 mol % of FL dyed lipid 1-oleoyl-2-{6-[(7-nitro-2-1,3-benzoxadiazol-4-yl)amino]hexanoyl}-sn-glycero-3-phosphocholine (NBD-PC). To predict the frequency-dependent dielectric behavior of SUVs, the CM factor was calculated by utilizing the walled cell model designed by Jones (Supplementary Note 16, Supplementary Table 5, and Supplementary Fig. 18), followed by experimental corroborations. As the applied $V_{pp}$ increased by 0.1 V every 10 s at a fixed frequency of 100 kHz, the SUVs began to be captured from $V_{th} = 1.76$ V ($n = 81$) with standard deviation of 0.23 V (indicated as an error bar in inset of Fig. 5a), and the amount captured increased accordingly with time and voltage amplitude (Fig. 5a). This low-voltage operation is one of the key prerequisites for the practical application of EVs on electric-biochips as it eliminates any possible scenarios of undesirable molecular phase transitions or thermal denaturation. Moreover, ACEO- and DEP-induced trapping of 50 nm SUVs to the center and edge of the VNEs were achieved at 10 kHz and 100 kHz, respectively, under a low voltage of 2 V over the millimeter-scale large area (Fig. 5b, c). From an analysis of the FL intensities on 400 unit cells, the variations of normalized FL intensities on each cell demonstrated uniformity and device robustness over a large area (Supplementary Note 17 and Supplementary Fig. 19). Additionally, the VNE patterns exhibited a high degree of design flexibility (emblems of 'KIST' and 'SNU'; Fig. 5d), demonstrating the applicability of various pattern designs over large areas (Fig. 5e and Supplementary Movie 2). This demonstrates the potential applications of our VNEs for various optical and electrical structures as well as high-throughput analysis, which has remained challenging on conventional HNE configurations.

**Trapping of Aβ₄₂ assemblies on VNE.** Finally, the device performance was demonstrated at the molecular level by using real biomolecules of amyloid-beta 1–42 (Aβ₄₂) peptides, which are the predominant Aβ species and the major cause of Alzheimer's disease (AD)[55]. Through self-assembly, Aβ₄₂ peptide monomers are aggregated and form ordered structures of oligomers, protofibrils, and fibrils[56]. Among them, the Aβ₄₂ oligomeric structure is more toxic than others because it binds to the neuronal synaptic membrane and causes a disruption of calcium homeostasis by forming Amyloid channels[57]. Despite the huge interest for its high association with the pathology of AD, studying the oligomeric Aβ₄₂ is challenging owing to its extremely small size, ultra-low concentration, structural instability, and heterogeneity[58]. Therefore, new methods to collect and dynamically relocate the Aβ₄₂ assemblies from low-concentrated solutions are highly desirable. To this end, a solution of 100 μM Aβ₄₂ peptides was

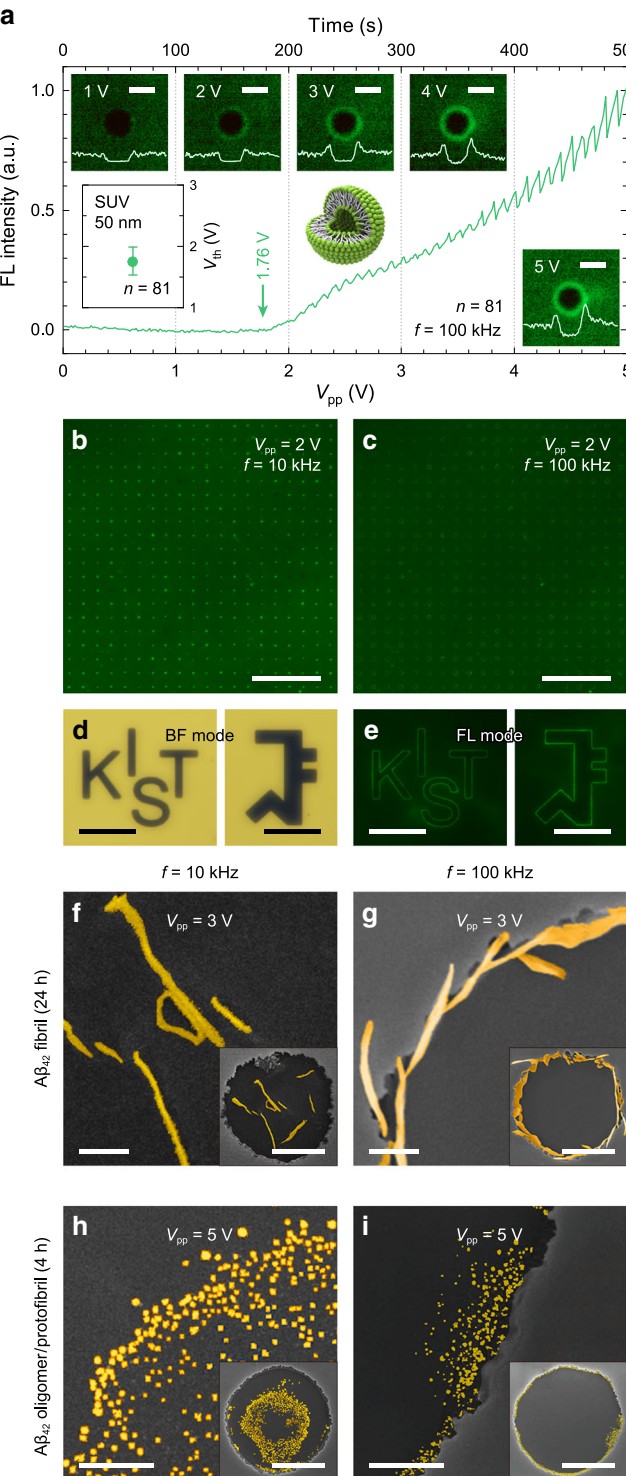

**Fig. 5 Trapping of nanoparticulate biomaterials on the VNE with large-area applicability and design flexibility. a** Averaged FL intensities and corresponding FL micrographs of captured 50-nm SUVs on 81 unit cells as a result of increasing $V_{pp}$ ($f = 100$ kHz) from 0 to 5 V by 0.1 V every 10 s; averaged trapping threshold of 1.76 V (inset). Scale bars, 10 μm. **b, c** FL micrographs of 50-nm SUVs trapped at center (**b**) and edge (**c**) of unit cells over large area under $f = 10$ and 100 kHz, respectively ($V_{pp} = 2$ V). Scale bars, 200 μm. **d, e** BF micrographs of VNE pre-defined with emblem patterns (KIST and SNU; **d**), and corresponding FL micrographs of 50-nm SUVs captured at edge of VNE electrode under $V_{pp} = 2$ V and $f = 100$ kHz (**e**). Scale bars, 40 μm. **f–i** SEM images of Aβ42 assemblies incubated for 24 h (**f, g**), and 4 h (**h, i**) trapped at center and edge of the VNE electrodes under AC frequencies of $f = 10$ kHz (**f, h**) and 100 kHz (**g, i**), respectively; $V_{pp} = 3$ V (**f, g**) and 5 V (**h, i**). Scale bars, 1 μm and 5 μm (inset).

the case of ultra-low concentration, additional FL photobleaching prohibited the reliable monitoring of captured Aβ42 through FL microscopy. Thus, SEM observation was performed for cross-validation. The fibrous Aβ42 peptides were observed at the center of the pattern at 10 kHz (Fig. 5f), while they were concentrated on the nanogap at 100 kHz (Fig. 5g). In contrast, spherical Aβ42 peptides were found at the center (Fig. 5h; 10 kHz) and edge of the pattern (Fig. 5i; 100 kHz) when exposing the 4 h-incubated Aβ42 solution to a higher AC bias of 5 V. To characterize the secondary structures of the 4 h-incubated Aβ42, a circular dichroism spectrometer was used. The results revealed 13.4% helix, 30.1% β-strand, 8.6% turns, and 47.8% other structures (Supplementary Note 19 and Supplementary Fig. 22), indicating that 4 h-incubated Aβ42 peptides are mostly in the form of oligomers and protofibrils[59]. Additional size validation and analysis were conducted using dynamic light scattering measurements and SEM spectroscopy, which represent the diameter variation of particulate Aβ42 from 10 to 60 nm, with the majority found to be 20 and 40 nm (well-matched with the reported size of oligomeric and protofibrous Aβ42)[60]. Because the 5 V application on VNE is regarded to generate $\Delta T < 0.01$ K (Supplementary Note 7 and Supplementary Fig. 5), any possible damage of the captured Aβ42 by Joule heating was ruled out. The results clearly demonstrate that our VNE can capture and relocate the Aβ42 aggregates down to the oligomeric and protofibrous levels without damage.

## Discussion

Although low-voltage NP trapping on HNE was intensively exploited through atomic layer lithography[27,28,61], the intrinsic nature of the horizontal configuration of its electrodes limited its practical application for wide spectral ranges of surface-sensitive optical sensors possessing maximized sensitivities. In comparison, three major advantages can be observed using the vertical configuration for NP manipulation. First, the VNE generates stronger DEP forces than HNE. The dielectrophoretic ability of the proposed VNE was investigated for the trapping of 200 nm PS beads with insulating thickness of 100 nm and minimum voltage of 0.33 V (Fig. 4). To the best of our knowledge, the performance observed is comparable to that of 'the world's sharpest DEP tweezer' for capturing 190 nm PS beads in the presence of 0.5 V, as reported by Barik et al.[27,62]. This evidently implies that the reduction of insulating thickness in the VNE platform will be significantly advantageous for realizing high-production yields as well as for trapping extremely small NPs, which have yet to be captured by conventional HNEs (Fig. 1). Second, based on the understanding of frequency dependency of electrokinetics, the proposed VNE helps control the local position of NPs. Although most of HNEs have focused on capturing target NPs along the restricted nanogap regions, the proposed VNEs not only capture NPs along the nanogap regions in various shapes but dynamically

incubated separately with different durations (4 h vs 24 h) in deionized water at 37 °C and stained with thioflavin-T for visualization, and the Aβ42 solution was diluted into 1 nM to emulate the actual ultra-low concentration of Aβ42 peptide. Prior to the experiments, the CM factors of Aβ42 peptides were calculated by using dielectric models depending on molecular shapes and sizes (Supplementary Note 18, Supplementary Table 6, and Supplementary Fig. 20). For the 24 h-incubated samples without the delusion process, the FL micrographs clearly exhibit fibrous Aβ42 peptides confined at the edge of the VNE patterns under an AC bias of 3 V at 100 kHz (Supplementary Fig. 21). However, in

control movements and local the position of NPs depending on the applied frequency (Fig. 3). Third, the VNE is advantageous for ease of fabrication, thereby achieving large-area scalability as well as reliability for massive sensor applications. Conventionally, HNEs have been developed through high-cost fabrication techniques such as e-beam lithography and focused ion beam. Although alternative methods were recently introduced, they still suffer from reliability issues and complicated nanofabrication processes[63]. Finally, owing to the high degree of design flexibility as well as the massive and simple fabrication, the proposed VNE could offer opportunities to develop an advanced optoelectronic sensor system by converging with various optical metasurface designs. Because the proposed VNE guarantees a high degree of design flexibility, the adoption of various optical nanostructures and sequential concentration of target NPs into the area of sensing hotspot will create tremendous opportunities to realize practical analytic methods, including surface-enhanced Raman spectroscopy (SERS)[64], surface-enhanced infrared absorption (SEIRA)[65], surface plasmon resonance[66], and terahertz sensing based on metasurfaces[67,68].

## Methods

**Voltage applications and data analysis**. After mixing a poly(dimethylsiloxane) (PDMS) elastomer and a curing agent (Sylgard 184, Dow Corning) at a ratio of 10:1 with a total weight of 3 g, the mixture was poured and evenly spread on a petri dish (size of 120 mm × 120 mm) and cured on a hotplate (80 °C for 3 h) to achieve a 200-μm-thick PDMS film. Then, the PDMS film was punched using a 4-mm-diameter biopsy punch (MIL3334HH, Miltex) and placed on the periphery of the VNE patterns. A solution volume of 10 μL was dropped on the VNE array and covered with a glass slip to prevent water evaporation during voltage application. The customized software was designed using LabVIEW 2018 (National Instruments) to provide systematically controlled AC signals from a waveform generator (SDG2122X, SIGLENT). In this process, the FL and BF images of particulate materials were monitored using a FL microscopic system (LV100ND, Nikon) and a confocal microscopic system (Eclipse Ni-E, Nikon) and then analyzed using image processing software (ImageJ, National Institutes of Health, USA).

**SEM observation**. Deposition of 2-nm-thick platinum on the samples was carried out using an ion sputter coater (E-1045, Hitachi) prior to observation with a SEM (Nova NanoSEM 200, FEI). An accelerating voltage of 10 kV and current of 140 μA was applied. For high-resolution SEM operation, a through-the-lens detector was used.

**Reporting summary**. Further information on research design is available in the Nature Research Reporting Summary linked to this article.

## Data availability

The data that support the findings of this study are available from the corresponding author upon reasonable request.

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

## Acknowledgements

This work was supported in part by KIST intramural grants (Nos. 2E30506, 2V07880 and 2E30140). Y.-S.R. acknowledges the support from Samsung Research Funding & Incubation Center of Samsung Electronics under Project Number SRFC-TE19303-01; S.-D.L., Brain Korea 21 Project funded by Ministry of Education of Korea; M.S., National Research Foundation of Korea (NRF) Global Frontier Program (No. CAMM-2019M3A6B3030638); Y.T.B., National Research Council of Science & Technology (NST) funded by the Korea government Ministry of Science and ICT (No. QLT-CRC-18-02-KICT); and S.-Y.L., the National Research Foundation of Korea (NRF) grant funded by the Korea government Ministry of Science and ICT (No. 2017R1C1B2003585).

## Author contributions

E.-S.Y. and Y.-S.R. designed and performed all the experiments, analyzed the data, and wrote the initial draft of the manuscript. E.-S.Y. and J.L. fabricated the devices. E.-S.Y., C.K., M.S., J.H.K., Y.T.B., S.-C.P., and S.-Y.L. performed the simulations and contributed to the data analysis. H.L. prepared the Aβ$_{42}$ sample and performed the CD and DLS measurements. S.-M.L. and J.K. prepared yeast cells and *Bacillus subtilis* spores. T.K. performed the SEM measurements. S.-D.L. and Y.-S.R. provided scientific and technical guidance, performed the interpretation of the data, and finalized the manuscript. All authors agreed on its final content.

## Competing interests

The authors declare no competing interests.
