## [Peer Review File · Nature Communications]

Reviewers' comments first round:

Reviewer #1 (Remarks to the Author):

Report on manuscript NCOMMS-19-1124856

"Precise capture and dynamic relocation of nanoparticulate biomolecules through the dielectrophoretic enhancement by vertical nanogap"

The problem of capture and relocation of micro- and nano-sized particles is addressed here. The authors propose a novel geometry of electro-hydrodynamic cell, able to capture and size screen particles more effectively than already existing systems. After a swift evaluation of the advantages and drawbacks of the new vertically layered system (VNE) from an electrical standpoint, results of Comsol simulations are reported concerning (i) the map of the electric field – or equivalently that of the gradient of the square of its norm (ii) the movement/dynamics of particles in the vicinity of the patterned surface. Then, experimental results regarding possible selective capture of particles (based on size) are reported. First, polystyrene nano particles are considered. Capture of Phospholipid unilamellar vesicle is reported in the next section, and the article ends with a section dedicated to the trapping of A β 42 biomolecules.

Despite a good initial idea and a substantial and probably challenging work in terms of manufacturing, the article suffers severe flaws concerning the structure and the contents, which in my opinion makes it unsuitable for publication in Nature Comm. The main problem is that there is almost no interpretation of the results. Even if the objective is not a theoretical analysis of the observations, the authors should provide the reader with some clues based on relevant scaling laws regarding the physical processes involved. In that regard, a dimensionless analysis of the particle dynamics would have greatly helped.

1. First, the message of the paper sounds blurry. I understand that the author performed a gradual testing of their device from the largest objects to the smallest, but an extensive dedicated study focused on one of these types would have been more convincing since it would have allowed a more thorough analysis of the processes at stake, and also a more exhaustive presentation of the results. Here, the reader is drowned under a bunch of results which have no real connecting thread.
2. Second, the device itself (without particles) would have deserved a dedicated analysis. The rectified electro-osmotic/electro-hydrodynamic flow has to be characterised more accurately. Because the particles are not passive, I understand that micro-PIV is probably irrelevant, but an hydrodynamical numerical analysis of the cell using Comsol (Stokes mode) and Electro-osmotic slip velocity at the boundaries is indispensable before going further. I reckon that this is partially done in the supplementary materials, but this should be part of the main article. If this is not possible, it means that the format of Nature Comm is not suitable for the purpose.
3. No rigorous equation is provided for the dynamics of the particle. Technical details on the forces at stake are provided in the supplementary materials and if I get it right, the authors write the total force as the sum of a dielectro-phoretic force and an electro-osmotic, which is the drag force induced by the hydrodynamic flow, itself coming from electro-osmotic slip velocity at the walls... Hard to tell from the body of the article. Supplementary materials seem to suggest that this is the case. In brief, the author should provide a) the rigorous form of what they call the electro-osmotic force b) a comprehensive equation for the dynamics of the particle (even if the main purpose of the article is not theoretical) - is inertia relevant (probably not), what is the Stokes number for these polystyrene particles? Etc.
4. Based on comments made in point 3, another question can be raised concerning the applicability of the so-called model to three species which are physically and chemically very different. Comments should be made on the relevancy of the model towards each type of particles. Possible local surface charges should be commented.

5. Again, related to the points previously listed, no trapping scenario is proposed. Most of the time, trapping results from the balance between two forces, but could also result from the localisation of the particle at a minimum of potential related to one single particular force (as it does for electrostatic trapping). A more thorough and rigorous analysis of the physics would help regarding this.

6. Brownian motion and molecular diffusion are kept silent throughout the paper. I agree that they are probably negligible in most cases, but quantitative comments must be made on this specific point, especially if the particles' Stokes numbers are small.

7. Under AC fields, particles will acquire an electric dipole moment (especially at high frequency, where Debye layer cannot screen the charges), which will trigger dipole interactions and subsequent interaction forces (attractive or repulsive depending on the relative positions of each particles, but most of the time attractive). Again, no comment is made about this potentially cluster generating process.

8. There are big gaps in the literature. Articles by A. Ajdari, Castellanos and Squires, among others, should be mentioned.

Reviewer #2 (Remarks to the Author):

REVIEW of NCOMMS-19-1124856

This manuscript describes a novel approach to generating strong electric field gradients using very low voltages, enabling the electrokinetic manipulation of nanoparticles (NPs) (synthetic and biological). This is achieved by the fabrication of a device with a vertical nanogap electrode (VNE) created by sandwiching a dielectric layer between two electrodes: ITO (40 nm bottom electrode) PVP (100 nm dielectric layer) and Au (40 nm top electrode). This vertical configuration enables a flexible configuration of trapping locations or geometries. In this work, the authors showed an array of 10 μ m circular wells and an array with "KIST" emblem pattern with the VNEs to trap and manipulate NPs.

The authors demonstrate the potential for VNEs to manipulate NPs of different sizes (1000, 200, 100 and 50 nm PS spheres, A β 42 peptides, 50 nm small unilamellar vesicles, and *B. subtilis* spores) and achieve selective trapping of particles by adjusting the AC electric potential and frequency to induce different regimes of electrokinetic behavior: 1-10 kHz dominated by AC electroosmosis (ACOE), 0.1-1 MHz dominated by positive dielectrophoresis (pDEP), and >10 MHz dominated by negative dielectrophoresis (nDEP).

This study also provides a comprehensive insight of the difference forces and phenomena present in the VNE system and how these can be used to predict particle behavior and optimize the VNEs design. The authors considered ACEO, DEP, Joule heating, electrothermal effects and gravitational force into the particle velocity and present simulations that are used to predict particle behavior and justify the experimental conditions employed to manipulate the NPs.

As the authors mentioned in their discussion of previous studies using nanogap electrodes, this concept is not entirely new, although this study is the first one showing a vertical nanogap geometry with the flexibility to adapt a variety of surface designs over a large area, providing a platform that would enable optical detection over large areas.

I believe this work will be of great interest, since this approach show a novel fabrication design to exploit the VNEs advantages with a platform that allows for the fabrication of VNEs over large areas. Is still unclear to what degree the fabrication methods allow for consistent VNEs to be assembled over these large areas. Nonetheless, the capacity to generate such high electric field gradients with low potentials in a stable (no significant changes in temperature and solution due to direct contact with the electrodes) system, is something that a lot of researchers are looking into in order to manipulate NPs for a wide range of applications without the need of tagging or pretreating the samples of interest. The VNEs described in this work will shift the idea of

conventional methods of generating electric field gradients, from horizontal electrodes, 3D carbon or coated electrodes, insulating structures, to these vertical nanogaps.

The manuscript in its present form needs to be proofread (I have included a highlighted copy of the manuscript and supplementary information), to improve its layout and of the supplementary information. There is no Introduction/Background section header, and the figures of the supplementary Information are either miss-referenced and positioned in an order that makes the reading hard going back and forth.

The authors should include repetitions of their experiments with the different (or a subset of) NPs and provide a standard deviation or error for the normalized intensity or threshold potential. Additionally, I would be very interested in seeing the device robustness in terms of the fabrication process (i.e. evaluate the large area performance of the device by measuring FL intensity at different regions of the VNEs array).

The authors can also mention how they prepared the 200 μm layer of PDMS (spin coating, fixed volume over fixed area) in the Supplementary Information.

RESPONSE TO DECISION LETTER

MANUSCRIPT INFO

Manuscript ID: Nature Communications Manuscript NCOMMS-19-1124856

Title: "Precise capture and dynamic relocation of nanoparticulate biomolecules through dielectrophoretic enhancement by a vertical nanogap architecture"

CONTENTS

Point-by-point response to comments from reviewer # 1: p. 3

Point-by-point response to comments from reviewer # 2: p. 28

Additional Changes: p. 38

FORMATTING KEY

Reviewers' comments — ***Bold Italic Font***

Author Response— Regular Font

Author Actions— *Italic*

Text Changed — Normal Underlined Font

1. POINT-BY-POINT RESPONSE TO COMMENTS FROM REVIEWER # 1

1.1. First, the message of the paper sounds blurry. I understand that the author performed a gradual testing of their device from the largest objects to the smallest, but an extensive dedicated study focused on one of these types would have been more convincing since it would have allowed a more thorough analysis of the processes at stake, and also a more exhaustive presentation of the results. Here, the reader is drowned under a bunch of results which have no real connecting thread.

The authors agree with reviewer #1's critics that the previous manuscript was lacking of exhaustive analysis on particle dynamics as well as device performance generating ACEO despite its gradual testing of particle manipulation from the largest object to the smallest. Especially, it lacks information related to the electro-osmotic slip velocity and dielectrophoretic phenomena with respect to a variety of particulate properties (surface charge, shape etc.) and interactions with the captured particles.

As reviewer #1 pointed out, the authors found that the previous manuscript could mislead our main intention of this work; Introducing applicability of our vertically-aligned nanogap electrodes (VNE) to wide range of nanoscale biomolecules based on overall interpretation, majorly assisted by fluidic flows (ACEO) and capturing forces (DEP). Now in present version, the authors have provided additional analysis explaining how and why those two major forces interplay to relocate the particles in the level of tens of nanoscale. The authors believe and hope that overall interpretation of our VNE device's performance through numerical analysis will highlight the novelty of this work and solve the issues that reviewer #1 pointed out. Thank you again for giving us opportunity to improve quality of our manuscript and providing advices to extend depth of the work.

Please, check the additional comments addressed one by one below.

1.2. Second, the device itself (without particles) would have deserved a dedicated analysis. The rectified electro-osmotic/electro-hydrodynamic flow has to be characterised more accurately. Because the particles are not passive, I understand that micro-PIV is probably irrelevant, but an hydrodynamical numerical analysis of the cell using Comsol (Stokes mode) and Electro-osmotic slip velocity at the boundaries is indispensable before going further. I reckon that this is partially done in the supplementary materials, but this should be part of the main article. If this is not possible, it means that the format of Nature Comm is not suitable for the purpose.

The authors agree with reviewer #1's opinion that more dedicated analysis on electro-osmotic flow is necessary. The authors have conducted additional simulations to characterize ACEO-driven local slip velocities at the boundaries of two electrodes as well as resultant bulk fluidic flow across the insulator of VNE, compared to the case of horizontally-aligned nanogap electrodes (HNE). In the course of additional simulations, the authors found out that asymmetric electrode configuration of VNE generates unbalanced slip velocities at each surface of two electrodes resulting in a pair of hydrodynamic bulky flows in different magnitudes and directions. In contrast to the symmetric HNE's equivalent slip velocities and micro-vortices, our VNE provides unique electrokinetic flows with asymmetric slip velocities and resultant asymmetric micro-vortices. This offers a clue to investigate particles' hydrodynamic flow-assisted behaviour at different applying frequencies (*i.e.*, particle mixing at 1 kHz and island-type particle relocalization at 10 kHz).

The Authors newly added figures and explanation about ACEO-induced bulk fluidic micro-vortices under 1 kHz (Figs. 1h and 1i in main manuscript) and 10 kHz (Fig. S6 in supplementary information) together with electro-osmotic slip velocities at the electrode surface of VNE (Figs. 1j-1l in main manuscript) and compared their performances with that of symmetric HNEs (Fig. S7 in supplementary information). Related comments on ACEO in Figs. 1 and 3 were changed accordingly.

FIGURE ADDED: Main Figs. 1h-1l (newly added).

FIGURE LEGEND REVISED:

“**h, i** Calculated distribution of ACEO flow on VNE (**h**) and HNE (**i**) with a gap distance of $d = 100$ nm under $V_{pp} = 1$ V and $f = 1$ kHz. **j-I** Calculated u_{slip} distribution at the electrode surfaces of VNE ($d = 100$ nm) as a function of f (**j**, $V_{pp} = 1$ V) with the schematic of u_{slip} (inset in **j**) and 3D illustrations of u_{slip} on electrode 1 (**k**, top electrode) and 2 (**l**, bottom electrode).”

SUBTITLE REVISED: Page 4

“**Dielectrophoretic- / AC electro-osmotic characterization and fabrication of VNE**”

TEXT ADDED: Main manuscript Page 2

“The VNE array, which is designed to generate well-controlled flow dynamics accompanied with improved particle capture, ...”

TEXT ADDED: Main manuscript Page 5-6

“Together with DEP performance, ACEO-driven local slip velocities at the boundary of two electrodes and resultant bulk fluidic flow across the insulator of both VNE and HNE were explored. The simulations of hydrodynamic bulky flow over the each electrode revealed that VNE generates a pair of micro-vortices rotating in opposite directions with distinct magnitudes (**Fig. 1h**), in contrast to symmetric HNEs, which yield bulky flows of equivalent magnitude in an outward direction from the nanogap (**Fig. 1i**). As is well known, the slip velocity, u_{slip} , arising at the electrode surface from the interaction of the E-field and EDL, drives the bulky fluidic motions. We analysed u_{slip} to characterize ACEO behaviour as a function of both frequency and position on the VNE surface (**Figs. 1j-1l**). On the boundary of the VNE surface, u_{slip} reached a maximum at the edge became decreased away from the nanogap irrespective of the applied frequency (**Fig. 1j**). Although the simulation results show good correspondence with those of previous studies examining ACEO-assisted slip velocity above HNEs^{32,37-39}, the VNE offers unique hydrodynamic flows resulting in asymmetry in magnitudes of u_{slip} at each micro vortex (**Figs. 1k and 1l**). This imbalance of a faster u_{slip} on the top electrode than that on bottom electrode arises from the structural configuration of VNE, as previously demonstrated by the use of asymmetric horizontal electrode arrays⁴⁰⁻⁴² or non-planar ones⁴³⁻⁴⁵. Owing to this faster u_{slip} on the top electrode, the micro-vortex on the top electrode induces larger flow streamlines than those on the bottom electrode (**Fig. 1h**). Moreover, this imbalance is intensified when the disparity of each flow across the nanogap increases under different frequency conditions, and this hints at the frequency-dependent distinct dynamics of particles in combination with DEP (**Supplementary Section 8**).”

REFERENCE ADDED: Main references [37] and [39-45]

[37] Ramos, A., Morgan, H., Green, N. G. & Castellanos, A. AC electric-field-induced fluid flow in microelectrodes. *J. Colloid Interf. Sci.* **217**, 420-422 (1999).

[39] Olesen, L. H., Bruus, H. & Ajdari, A. Ac electrokinetic micropumps: The effect of geometrical confinement, faradaic current injection, and nonlinear surface capacitance. *Phys. Rev. E* **73**, 056313 (2006).

[40] Ajdari, A. Pumping liquids using asymmetric electrode arrays. *Phys. Rev. E* **61**, R45-R48 (2000).

[41] Brown, A. B. D., Smith, C. G. & Rennie, A. R. Pumping of water with ac electric fields applied to asymmetric pairs of microelectrodes. *Phys. Rev. E* **63**, 016305 (2000).

[42] Ramos, A., Gonzalez, A., Castellanos, A., Green, N. G. & Morgan, H. Pumping of liquids with ac voltages applied to asymmetric pairs of microelectrodes. *Phys. Rev. E* **67**, 056302 (2003).

[43] Urbanski, J. P., Thorsen, T., Levitan, J. A. & Bazant, M. Z. Fast ac electro-osmotic micropumps with nonplanar electrodes. *Appl. Phys. Lett.* **89**, 143508 (2006).

[44] Bazant, M. Z. & Ben, Y. Theoretical prediction of fast 3D AC electro-osmotic pumps. *Lab Chip* **6**, 1455-1461 (2006).

[45] Urbanski, J. P., Levitan, J. A., Burch, D. N., Thorsen, T. & Bazant, M. Z. The effect of step height on the performance of three-dimensional ac electro-osmotic microfluidic pumps. *J. Colloid Interf. Sci.* **309**, 332-341 (2007).

REFERENCE MOVED: Main references [38]

[38] Green, N. G., Ramos, A., González, A., Morgan, H. & Castellanos, A. Fluid flow induced by nonuniform ac electric fields in electrolytes on microelectrodes. III. Observation of streamlines and numerical simulation. *Phys. Rev. E* **66**, 026305 (2002). ← moved from [37] of previous version

TEXT ADDED: Page 7

“In regime I, pairs of asymmetric ACEO flows circulating above each surface of VNE were simulated both in 1 kHz and 10 kHz. Because ACEO is predominant in the low-frequency range of regime I, the movements of particles are dominantly determined by ACEO rather than DEP. Within regime I, distinct particle dynamics are also expected with respect to the applied frequencies owing to the varying dominant forces: ACEO is dominant over DEP at 1 kHz (Fig. 3a), whereas they are comparable at 10 kHz (Fig. 3b), as shown through simulations (Supplementary Section 4). In contrast, the formation of the EDL layer is weakened at high frequency as the alternation of electrical signals is too rapid for ions to follow; thus, particles are majorly governed by F_{DEP} (Figs. 3c-3e).”

TEXT ADDED: Page 8

“At 1 kHz (Fig. 3f), the particles were swirling over the single unit of VNE because they are significantly influenced by ACEO flows rather than the considerably weaker DEP force. However, at 10 kHz, suspended particles are migrated to the centre of the unit cell, where flows are converged to form a point of stagnation (Fig. 3g and Supplementary Sections 4). Unlike at 1 kHz, an application at 10 kHz generates both ACEO and DEP with comparable magnitudes. With an aid of attracting F_{DEP} , particles are brought into proximity with the VNE electrode surface against upward ACEO flow. Then, because the ACEO-induced slip velocity (pushing toward centre) is found to be stronger than the DEP force (attracting toward edges) at the collecting surface of the VNE, effective surface streams push particles toward the stagnation point where the potential energy is the lowest (Supplementary Section 12)⁵¹. In contrast, at 100 kHz (Fig. 3h) and 1 MHz (Fig. 3i) in regime II, the ring-shaped particle assembly was observed at the edge of nanogap, where the value of ∇E^2 is the largest and the potential energy is the lowest (Supplementary Section 12). Although ACEO flows still exist at 100 kHz, trapping at the stagnation deteriorates as a pair of asymmetric micro-vortices vanish into a single large flow and F_{DEP} becomes stronger than F_{ACEO} (Supplementary Sections 4). Instead, a long-range ACEO flow constantly conveys distant particles to the VNE, and the dominant F_{DEP} successfully snatches the moving NPs from the flow and accumulate them to the edge of VNEs at 100 kHz.”

FIGURE ADDED: Supplementary Figs. S6 and S7 (newly added).

Supplementary Information Page 13: Supplementary Sections 8

“**Fig. S6 ACEO-induced micro-vortices under 10 kHz. a, b** Calculated distribution of ACEO flow on VNE (a) and HNE (b) with a gap distance of $d = 100$ nm under $V_{pp} = 1$ V and $f = 10$ kHz.”

“**Fig. S7 Analysis of ACEO-induced slip velocity on HNE. a-c** Calculated slip velocity (u_{slip}) distribution at the electrode surfaces of HNE ($d = 100$ nm) as a function of f (a, $V_{pp} = 1$ V) with schematic of u_{slip} (inset in a) and 3D illustrations of u_{slip} on electrode 1 (b) and 2 (c).”

1.3. No rigorous equation is provided for the dynamics of the particle. Technical details on the forces at stake are provided in the supplementary materials and if I get it right, the authors write the total force as the sum of a dielectro-phoretic force and an electro-osmotic, which is the drag force induced by the hydrodynamic flow, itself coming from electro-osmotic slip velocity at the walls... Hard to tell from the body of the article. Supplementary materials seem to suggest that this is the case. In brief, the author should provide a) the rigorous form of what they call the electro-osmotic force b) a comprehensive equation for the dynamics of the particle (even if the main purpose of the article is not theoretical) - is inertia relevant (probably not), what is the Stokes number for these polystyrene particles? Etc.

The authors agree with reviewer #1's concern that total force described in previous manuscript is insufficient to express dynamics of suspended particles in the water environment. It is now revised by adding Langevin equation with various external phenomena including dielectrophoresis, AC electro-osmosis, electrothermal flow, gravitation, buoyancy, Brownian motion, and inter-particulate interactions. According to the calculation in Supplementary information, the authors found that the effects of electrothermal flow, gravitation, buoyancy, Brownian motion are negligible in our case. For low concentration of suspended particles floating apart each other, inter-particulate interactions are also negligible except for special cases (the special cases will be discussed in 1.7 below). Thus, the exerting forces on suspended particles can be expressed in two governing phenomena; DEP and ACEO. The detailed information of each exerting forces on a single particle is expressed in the newly revised main manuscript. Besides, F_{ACEO} was defined rigorously in terms of Stokes drag force and addressed in detail together with dimensionless analysis on Stokes numbers of particles (Stk). All the details related to this issue will be addressed below. Note the accordance that the authors addressed in the order of reviewer #1's question (b) and (a) in the main manuscript.

1.3.1. In brief, the author should provide ... b) a comprehensive equation for the dynamics of the particle (even if the main purpose of the article is not theoretical)"

In the main manuscript, the comprehensive dynamics of particles are given as Langevin equation with dielectrophoresis, AC electro-osmosis, and the others. After demonstrating that effects of the others are negligible (Supplementary Information), the solution of comprehensive particle dynamics is derived from Langevin equation and expressed in terms of two dominant forces; DEP and ACEO (main manuscript).

TEXT REVISED: Page 3-4

“To evaluate the competitive forces and resultant dynamics of a spherical particle in an aqueous environment, the Langevin equation of particle velocity (u_p) was employed³⁵:

$$m_p \frac{du_p}{dt} = F_{DEP} + F_{ACEO} + F_{ETF} + F_{grav} + F_{buoy} + \zeta(t) + \sum F_{i,j} \quad (3)$$

considering DEP force (F_{DEP}), Stokes drag force induced by ACEO flow (F_{ACEO}) and electrothermal flow (F_{ETF}), gravitational force (F_{grav}), buoyant force (F_{buoy}), random Brownian force ($\zeta(t)$), and interaction

between neighbouring particles ($F_{i,j}$; interaction force acting on the i -th particle owing to the j -th particles). Because Stokes drag forces are defined as $F = -6\pi\eta R(\mathbf{u}_p - \mathbf{u}_m)$, where \mathbf{u}_p and \mathbf{u}_m are the velocities of particles and fluidic flows, respectively, F_{ACEO} and F_{ETF} exerting on a single particle can be evaluated by inserting \mathbf{u}_{ACEO} and \mathbf{u}_{ETF} into \mathbf{u}_m , respectively³⁶. Considering the simulations (F_{ETF} , F_{DEP} , and F_{ACEO}), calculations (F_{grav} , F_{buoy} , and $\zeta(t)$), and low concentration (10 ppm) condition ($F_{i,j}$) on the VNE having pattern size (L) and periodicity (P) of $L = 10 \mu\text{m}$ and $P = 30 \mu\text{m}$ (Fig. 1b), F_{DEP} and F_{ACEO} dominantly determine the movement of 1- μm -diameter polystyrene (PS) particles, while the others are negligible (Supplementary Sections 1-5). Moreover, owing to the small Brownian timescale for relaxation of the particle ($\tau_B = m/6\pi\eta R \approx 6 \times 10^{-8}$ s), equation (3) can be simplified into terminal \mathbf{u}_p as³⁶

$$\mathbf{u}_p = \left(\mathbf{u}_{ACEO} + \frac{\mathbf{F}_{DEP}}{6\pi\eta R} \right) \left\{ 1 - e^{-(6\pi\eta R/m)t} \right\} \approx \mathbf{u}_{ACEO} + \frac{\mathbf{F}_{DEP}}{6\pi\eta R} \quad (4)$$

with the assumption that the initial velocity of the particle is 0.”

REFERENCE ADDED: Main references [36]

[36] Castellanos, A., Ramos, A., Gonzalez, A., Green, N. G. & Morgan, H. Electrohydrodynamics and dielectrophoresis in microsystems: Scaling laws. *J. Phys. D: Appl. Phys.* **36**, 2584 (2003).

TEXT/FIGURE REVISED & NEWLY ADDED: Supplementary sections 1-5

Simulation & calculation of dielectrophoresis, AC electro-osmosis, electrothermal flow, gravitation, buoyancy, and Brownian motion. (texts, figures, tables) are rearranged and added.

1.3.2. “In brief, the author should provide a) the rigorous form of what they call the electro-osmotic force”

The rigorous form of F_{ACEO} is given and expressed in the main manuscript and supplementary information in detail.

TEXT ADDED: Page 4

“Because Stokes drag forces are defined as $F = -6\pi\eta R(\mathbf{u}_p - \mathbf{u}_m)$, where \mathbf{u}_p and \mathbf{u}_m are the velocities of particles and fluidic flows, respectively, F_{ACEO} and F_{ETF} exerting on a single particle can be evaluated by inserting \mathbf{u}_{ACEO} and \mathbf{u}_{ETF} into \mathbf{u}_m , respectively³⁶.”

TEXT ADDED: Page 7

“Note that the electrohydrodynamic drag forces acting on the particle by ACEO were calculated using $F_{ACEO} = -6\pi\eta R(\mathbf{u}_p - \mathbf{u}_{ACEO})$, with assumptions that E-field had just been applied and that the particle was at rest prior to the AC application ($\mathbf{u}_p = 0$)⁴⁶. The DEP-induced velocity of particles (\mathbf{u}_{DEP}) was also calculated and simultaneously expressed in the form of $\mathbf{u}_{DEP} = \mathbf{F}_{DEP}/6\pi\eta R$ for direct comparison with \mathbf{u}_{ACEO} .”

REFERENCE MOVED: Main references [46]

[46] Oh, J., Hart, R., Capurro, J. & Noh, H. M. Comprehensive analysis of particle motion under non-uniform AC electric fields in a microchannel. *Lab Chip* **9**, 62-78 (2009). ← moved from [33] of previous version

TEXT MOVED/REVISED/ADDED: Supplementary Information Page 8

“Section 4: Simulation results of F_{DEP} , F_{ACEO} , and F_{ETF} (u_{DEP} , u_{ACEO} , and u_{ETF}).

For the quantitative comparison of simulation results, the electrohydrodynamic drag forces acting on the particle owing to ACEO (F_{ACEO}) and ETF (F_{ETF}) are calculated using $F = -6\pi\eta R(u_p - u_m)$, in which u_p and u_m represents particle velocity and fluid velocity induced by ACEO (u_{ACEO}) and ETF (u_{ETF}), respectively. For calculating F_{ACEO} and F_{ETF} , $u_p = 0$ was initially assumed, indicating that electric potential has just been applied, as mentioned in the main manuscript²². Meanwhile, DEP-induced particle velocity ($u_{DEP} = F_{DEP}/6\pi\eta R$) is also calculated and compared with u_{ACEO} and u_{ETF} . These results (Fig. S4) show that ETF is negligible (in the order of 10^{-18} N and 10^{-10} m/s) compared with DEP and ACEO (in the order of 10^{-18} N and 10^{-10} m/s).”

REFERENCE ADDED: Supplementary references [22]

[22] Oh, J., Hart, R., Capurro, J. & Noh, H. M. Comprehensive analysis of particle motion under non-uniform AC electric fields in a microchannel. *Lab Chip* **9**, 62-78 (2009).

1.3.3. *“is inertia relevant (probably not), what is the Stokes number for these polystyrene particles? Etc.”*

The importance of Stokes numbers on dynamics of particles is discussed and calculated in Supplementary Information. Also, the results and meanings are briefly mentioned in the revised main manuscript.

TEXT ADDED: Supplementary Information Page 11

“Section 6: Stokes numbers of particles.

The Stokes number (Stk) is the dimensionless number that characterizes the particle behaviour suspended in a fluid flow, which is defined as the ratio of characteristic time of suspended particle to that of the fluid²⁴

$$Stk = 2\rho_p u R^2 / 9\eta L. \quad (11)$$

A particle with $Stk \gg 1$ maintains its initial trajectory, without much deflection, instead of following the streamline of flow. On the contrary, for $Stk \ll 1$, the motion of the particles is dominated by the viscous force with negligible inertial effect, and therefore, is tightly coupled to the fluid streamlines. Since velocities arising from ACEO dynamics were calculated to be in the order of $\sim 10^{-4}$ m/s (Section 4 and Main Fig. 3), an $Stk \ll 1$ is calculated for the moving PS particles with various diameters used in our experiments. For example, Stk of PS particles with diameters of 1 μm , 200 nm, 100 nm, and 50 nm are $Stk \approx 7 \times 10^{-7}$, 3×10^{-8} , 7×10^{-9} , and 2×10^{-9} , respectively. Therefore, we conclude that suspended particles of our experiments move along the fluid streamlines, instead of detaching from the streamlines where flow abruptly changes.”

REFERENCE ADDED: Supplementary references [24]

[24] Liu, S.-J., Wei, H.-H., Hwang, S.-H. & Chang, H.-C. Dynamic particle trapping, release, and sorting by microvortices on a substrate. *Phys. Rev. E* **82**, 026308 (2010).

TEXT MOVED/REVISED/ADDED: Page 4

Calculated Stokes numbers with physical meaning is briefly commented.

“From a dimensionless analysis of particles that exhibit small Stokes numbers (**Supplementary Section 6**), the viscous fluidic effect is expected to dominate over the negligible inertial effect and the motions of particles will be coupled with flow. Therefore, the movements and positions of the suspended particles under various AC conditions can be anticipated by calculating two dominant electrokinetics of DEP and ACEO.”

1.4. Based on comments made in point 3, another question can be raised concerning the applicability of the so-called model to three species which are physically and chemically very different. Comments should be made on the relevancy of the model towards each type of particles. Possible local surface charges should be commented.

The authors are very pleased to improve our manuscript through adding more data which address the device performance in particle captures with different physicochemical properties. As reviewer #1 pointed out, dielectrophoretic response of particles with different physicochemical properties should be addressed for the bio- and nano-particulates in this article.

As a first step, the authors chose the appropriate dielectric models for particles with different structural characters and discussed in detail. By referring previous pioneering studies, Maxwell-Wagner-O’Konski (MWO) model (for PS particle, *B. Subtilis*, and Amyloid beta protofibrils/oligomers), multi-shell model (for spherical yeast cell), elongated multi-shell model (for spheroidal yeast cell), walled cell model (for vesicle), and elongated model (for Amyloid beta fibrils) were utilized based on their structural similarities. Based on these models, CM factor values were calculated as a function of applied frequency.

Taking a step further, the role of local surface charge on a sub-micron particle is also investigated together with its dielectrophoretic properties in the presence of applied voltages. Since the ionic currents and its convection within the EDL play an important role in polarization and induced dipole of nanoscale particle with its size comparable to EDL thickness, MWO theory was utilized to characterize these in terms of surface conductivity.

1.4.1. “Based on comments made in point 3, another question can be raised concerning the applicability of the so-called model to three species which are physically and chemically very different. Comments should be made on the relevancy of the model towards each type of particles.”

Dielectric models for different types of particles including PS particles, B. subtilis, SUVs, A β ₄₂ fibrils, A β ₄₂ oligomer/protofibrils were addressed, and their CM factors were calculated respectively. During the revision process, dielectroporetic and trapping characteristics of yeast cells were newly included and calculated in identical manner. The detailed explanation of each models, references of physical parameters, and calculation of CM factors were included in Supplementary Information and briefly mentioned in Main Manuscript.

TEXT REVISED: Supplementary Information Page 16-17: Supplementary Section 10

Detailed explanations on dielectric model, and CM factor calculation of PS particles. Based on MWO theory.

“Calculation of CM factor of PS micro/nanoparticles. The frequency-dependent CM factor, $f_{CM}(\omega)$, can be calculated using the following equation:

$$f_{CM}(\omega) = \frac{\epsilon_p^*(\omega) - \epsilon_m^*(\omega)}{\epsilon_p^*(\omega) + 2\epsilon_m^*(\omega)} \quad (12)$$

where $\epsilon_p^*(\omega)$ and $\epsilon_m^*(\omega)$ denote the complex permittivity of the particle and medium, respectively. Each complex permittivity is expressed as,

$$\varepsilon_p^*(\omega) = \varepsilon_p^* - j \frac{\sigma_p}{\omega} \quad \text{and} \quad \varepsilon_p^*(\omega) = \varepsilon_p^* - j \frac{\sigma_p}{\omega} \quad (13)$$

in which ε_p , ε_m , σ_p , σ_m are the permittivities and conductivities of the particle and medium, respectively ($j = \sqrt{-1}$). Equations (12) and (13) indicate that the polarizing behaviour is majorly governed by the material properties of the particles and medium conditions. Additionally, the local surface charge of the particles can be evaluated by adopting the Maxwell-Wagner-O'Konski (MWO) model, which is given by²⁵,

$$\sigma_p = \sigma_{\text{bulk}} + 2K_{\text{surf}}/R \quad (14)$$

where σ_{bulk} and K_{surf} denote the bulk conductivity and surface conductance of the particles, respectively. Applying the parameters specified in Table S2 to equations (12-14), the real parts of the CM factor ($\text{Re}[f_{\text{CM}}(\omega)]$) were successfully calculated using their effective conductivities of $\sigma_p = 114 \mu\text{S/cm}$, $570 \mu\text{S/cm}$, $1140 \mu\text{S/cm}$, and $2280 \mu\text{S/cm}$ for particles with diameters of 1 μm , 200 nm, 100 nm, and 50 nm, respectively (Fig. S9).”

TABLE ADDED: Supplementary Table S2. Parameters for calculation of CM factor of PS particles

REFERENCE REMOVED: Supplementary references ([26] in previous version)

FIGURE REVISED: Supplementary Fig. S9. CM factor of PS particles are newly calculated and plotted in the revised SI manuscript.

“**Fig. S9** Calculated CM factors of PS particles. Calculated $\text{Re}[f_{\text{CM}}(\omega)]$ of PS particles as a function of frequency; diameters of 1 μm (black), 200 nm (blue), 150 nm (green), and 50 nm (red). Note that corresponding crossover frequencies (f_c) of PS particles were estimated to be 1.8 MHz (1 μm), 12 MHz (200 nm), 18 MHz (100 nm), and 37 MHz (50 nm), respectively.”

TEXT ADDED: Supplementary Information Page 22-24: Supplementary Section 14

Detailed explanations about sample preparation, dielectric model, and CM factor calculation of yeast cells are added. The authors have applied multi-shell model for spherical-shaped cell and elongated multi-shell model for spheroidal-shaped cell.

“Section 14: Trapping of yeast on VNE.

Preparation of yeast cells. To prepare the yeast cells, *Yarrowia lipolytica* ATCC MYA-2613 strain was pre-inoculated into yeast synthetic complete (YSC) medium containing 6.7 g/L yeast nitrogen base (YNB), 20 g/L glucose, and complete supplement mixture (MP Biomedicals, Solon, OH, USA), and cultivated for 24 h at 28 °C. The cells were then inoculated into a 250 ml flask containing 25 ml of the respective medium, and cultivated at 28 °C for 144 h with orbital shaking at 200 rpm. The elongated cells were prepared by exposing spherical cells to stress condition with DI water for 3 h.

Calculation of CM factor of yeast Cells. To simplify the complexity of the biomaterial structure, the multi-shell model was adopted to evaluate the effective dielectric response of spherical yeast cells²⁹⁻³¹. Based on the two-shell model consisting of three distinct regions of highly conductive cytoplasm, ultrathin lossy lipid membrane, and cell wall (**Fig. S13a**), the CM factor is calculated using homogeneous effective permittivity of the model. The CM factor is given as follows,

$$\varepsilon_p^*(\omega) = \varepsilon_w^*(\omega) \frac{(R_3/R_2)^3 + 2 \left[\frac{\{\varepsilon_2^*(\omega) - \varepsilon_w^*(\omega)\}}{\{\varepsilon_2^*(\omega) + 2\varepsilon_w^*(\omega)\}} \right]}{(R_3/R_2)^3 - \left[\frac{\{\varepsilon_2^*(\omega) - \varepsilon_w^*(\omega)\}}{\{\varepsilon_2^*(\omega) + 2\varepsilon_w^*(\omega)\}} \right]} \quad (15)$$

and the parameters are given by,

$$\varepsilon_2^*(\omega) = \varepsilon_{LB}^*(\omega) \frac{(R_2/R_1)^3 + 2 \left[\frac{\{\varepsilon_c^*(\omega) - \varepsilon_{LB}^*(\omega)\}}{\{\varepsilon_c^*(\omega) + 2\varepsilon_{LB}^*(\omega)\}} \right]}{(R_2/R_1)^3 - \left[\frac{\{\varepsilon_c^*(\omega) - \varepsilon_{LB}^*(\omega)\}}{\{\varepsilon_c^*(\omega) + 2\varepsilon_{LB}^*(\omega)\}} \right]} \quad (16)$$

$\varepsilon_w^*(\omega) = \varepsilon_w - j(\sigma_w/\omega)$, $\varepsilon_{LB}^*(\omega) = \varepsilon_{LB} - j(\sigma_{LB}/\omega)$, $\varepsilon_c^*(\omega) = \varepsilon_c - j(\sigma_c/\omega)$ $R_3 = R$, $R_2 = R_3 - t_w$, and $R_1 = R_2 - t_{LB}$, respectively.

However, for elongated yeast cells, the spheroidal shell model with a long axis radius of R_y and short axis radii of $R_x = R_z = R_y$ was utilized (**Fig. S13b**)^{32,33}. In this model, the CM factor is given by³³⁻³⁵

$$f_{CM}(\omega) = \frac{\varepsilon_p^*(\omega) - \varepsilon_m^*(\omega)}{\left\{ \varepsilon_p^*(\omega) - \varepsilon_m^*(\omega) \right\} A + \varepsilon_m^*(\omega)} \quad (17)$$

in which A denotes the depolarizing factor. Assuming that the E-field is parallel to the major x -axis for the ellipsoid of $R_y > R_x = R_z$, A is given by^{34,35}

$$A = -\frac{R_y^2}{2R_x^2 e^3} \left(2e - \ln \frac{1+e}{1-e} \right) \quad (18)$$

where $e = [1 - (R_x/R_y)^2]^{1/2}$. While $\text{Re}[f_{CM}(\omega)]$ in the spherical model is bounded within the range of $-0.5 < \text{Re}[f_{CM}(\omega)] < 1$, that of the spheroid model presents much greater $\text{Re}[f_{CM}(\omega)]$ values, indicating that the DEP force acting on the elongated particle is much stronger than that on the spherical ones (**Fig. S13c**). Since the attractive DEP force is stronger in the direction of the long-axis of spheroid, most elongated yeasts cells are aligned in the radial direction (**Fig. S14**)."

TABLE ADDED: Supplementary Table S3. Parameters for calculation of CM factor of yeast cells.

FIGURE ADDED: Supplementary Figs. S13 & S14. Dielectric model and CM factor of yeast cells & Trapping experiments of yeast cells.

“Fig. S13 Dielectric model and calculated CM factor of yeast cells. a-c Schematic illustrations of dielectric models of the yeast cell based on the two-shell model (a), and the ellipsoid shell model (b). Calculated $Re[f_{CM}(\omega)]$ of spherical- (red) and elongated yeast cells (blue) as a function of frequency (c). Note that f_r values of spherical and elongated yeast cells were estimated to be 35 MHz and 28 MHz, respectively.”

“Fig. S14 Trapping of yeast cells on VNE. a, b BF microscopic images of spherical- (a) and elongated yeast cells (b) trapped at the edge of the VNE electrodes under AC frequencies of $V_{pp} = 0.8$ V and $f = 100$ kHz. Scale bars: 20 μm . c SEM image of trapped elongated yeast cells. Scale bar: 5 μm .”

REFERENCE ADDED: Supplementary references [29-36]

- [29] Irimajiri, A., Hanai, T. & Inouye, A. A dielectric theory of “multi-stratified shell” model with its application to a lymphoma cell. *J. Theor. Biol.* **78**, 251-269 (1979).
- [30] Raicu, V., Raicu, G. & Turcu, G. Dielectric properties of yeast cells as simulated by the two-shell model. *Biochim. Biophys. Acta, Bioenerg.* **1274**, 143-148 (1996).
- [31] Turcu, I. & Lucaci, C. M. Dielectrophoresis: A spherical shell model. *J. Phys. A: Math. Gen.* **22**, 985-993 (1989).
- [32] Asami, K., Hanai, T. & Koizumi, N. Dielectric approach to suspensions of ellipsoidal particles covered with a shell in particular reference to biological cells. *Jpn. J. Appl. Phys.* **19**, 359-365 (1980).
- [33] Kakutani, T., Shibatani, S. & Sugai, M. Electrorotation of non-spherical cells: Theory for ellipsoidal cells with an arbitrary number of shells. *Bioelectrochem. Bioenerg.* **31**, 131-145 (1993).
- [34] Morgan, H. & Green, N. G. Dielectrophoretic manipulation of rod-shaped viral particles. *J. Electrostat.* **42**, 279-293 (1997).
- [35] Stratton, J. A. *Electromagnetic Theory*. (McGraw-Hill, New York, 1941).
- [36] Patel, S. *et al.* Microfluidic separation of live and dead yeast cells using reservoir-based dielectrophoresis. *Biomicrofluidics* **6**, 34102-34102 (2012).

TEXT ADDED: Supplementary Information Page 25-26: Supplementary Section 15

Detailed explanations on dielectric model, and CM factor calculation of B. subtilis. Based on MWO theory.

“Calculation of CM factor of B. subtilis. The CM factor of B. subtilis was calculated using MWO theory, simplifying the complexity of the biomaterial structure into a homogeneous dielectric solid (Fig. S15). Note that surface conductance was assumed from those of virus particles³⁹⁻⁴¹.”

TABLE ADDED: Supplementary Table S4. Parameters for calculation of CM factor of B. subtilis.

FIGURE ADDED: Supplementary Fig. S15. CM factor of B. subtilis.

“Fig. S15 Calculated CM factors of B. subtilis. Calculated $\text{Re}[f_{\text{CM}}(\omega)]$ of B. subtilis as a function of frequency. The f_c of B. subtilis was estimated to be 6.7 MHz.”

REFERENCE ADDED: Supplementary references [39-43]

[39] Hughes, M. P., Morgan, H. & Rixon, F. J. Dielectrophoretic manipulation and characterization of herpes simplex virus-1 capsids. *Eur. Biophys. J.* **30**, 268-272 (2001).

[40] Hughes, M. P., Morgan, H. & Rixon, F. J. Measuring the dielectric properties of herpes simplex virus type 1 virions with dielectrophoresis. *Biochim. Biophys. Acta, Gen. Subj.* **1571**, 1-8 (2002).

[41] Nakano, M., Ding, Z. & Suehiro, J. Dielectrophoresis and dielectrophoretic impedance detection of adenovirus and rotavirus. *Jpn. J. Appl. Phys.* **55**, 017001 (2015).

[42] Kell, D. B. & Harris, C. M. On the dielectrically observable consequences of the diffusional motions of lipids and proteins in membranes. *Eur. Biophys. J.* **12**, 181-197 (1985).

[43] Yu, A. C. S., Loo, J. F. C., Yu, S., Kong, S. K. & Chan, T.-F. Monitoring bacterial growth using tunable resistive pulse sensing with a pore-based technique. *Appl. Microbiol. Biotechnol.* **98**, 855-862 (2014).

TEXT ADDED: Supplementary Information Page 28-29: Supplementary Section 17

Detailed explanations on dielectric model, and CM factor calculation of small unilamellar vesicle (SUV) using walled-cell model of Jones.

“Calculation of CM factor of SUV. To evaluate induced dipole polarization of SUVs, the walled cell model of Jones, which consists of three different regions of cell interior, cell membrane and the wall, was adopted (Fig. S18a)⁴⁵. By replacing the wall and cytoplasm of the model with a lossy lipid membrane shell of finite thickness and an aqueous medium, respectively, the complex permittivity of the dielectric model ($\epsilon_p^*(\omega)$) is given by.

$$\varepsilon_p^*(\omega) = \varepsilon_w^*(\omega) \frac{(R_2/R_1)^3 + 2 \left[\frac{\{\varepsilon_c^*(\omega) - \varepsilon_{LB}^*(\omega)\}}{\{\varepsilon_c^*(\omega) + 2\varepsilon_{LB}^*(\omega)\}} \right]}{(R_2/R_1)^3 - \left[\frac{\{\varepsilon_c^*(\omega) - \varepsilon_{LB}^*(\omega)\}}{\{\varepsilon_c^*(\omega) + 2\varepsilon_{LB}^*(\omega)\}} \right]} \quad (19)$$

where each parameter is defined as, $\varepsilon_{im}^*(\omega) = \varepsilon_{im} - j(\sigma_{im}/\omega)$, $\varepsilon_{LB}^*(\omega) = \varepsilon_{LB} - j(\sigma_{LB}/\omega)$, and

$$\varepsilon_c^*(\omega) = \frac{c_{LB} R_1 \varepsilon_{im}^*(\omega)}{c_{LB} R_1 + \varepsilon_{im}^*(\omega)} \quad (20)$$

with their values specified in **Table S5**. By substituting $\varepsilon_p^*(\omega)$ of equation (12) into that of equation (19), the CM factor of SUV was predicted as a function of the applied frequency (**Fig. S18b**).

FIGURE ADDED: Supplementary Fig. S18. Dielectric model and CM factor of SUVs.

“Fig. S18 Dielectric model and calculated CM factor of SUV. a, b Schematic illustration of the dielectric model of SUV based on the walled cell model (**a**; Reproduced from [45]) and calculated $\text{Re}[f_{CM}(\omega)]$ of SUV as a function of frequency (**b**). The f_z of SUV was estimated to be 650 kHz.”

TABLE ADDED: Supplementary Table S5. Parameters for calculation of CM factor of SUVs.

REFERENCE ADDED: Supplementary references [45-54]

[45] Jones, T. B. *Electromechanics of Particles*. (Cambridge University Press, Cambridge, 1995).

[46] Yoon, T.-Y. *et al.* Topographic control of lipid-raft reconstitution in model membranes. *Nat. Mater.* **5**, 281-285 (2006).

[47] Ryu, Y.-S. *et al.* Reconstituting ring-rafts in bud-mimicking topography of model membranes. *Nat. Commun.* **5**, 4507 (2014).

[48] Ryu, Y.-S. *et al.* Lipid membrane deformation accompanied by disk-to-ring shape transition of cholesterol-rich domains. *J. Am. Chem. Soc.* **137**, 8692-8695 (2015).

[49] Ryu, Y.-S. *et al.* Curvature elasticity-driven leaflet asymmetry and interleaflet raft coupling in supported membranes. *Adv. Mater. Interfaces* **5**, 1801290 (2018).

[50] Ryu, Y.-S. *et al.* Kinetics of lipid raft formation at lipid monolayer-bilayer junction probed by surface plasmon resonance. *Biosens. Bioelectron.* **142**, 111568 (2019).

[51] Lim, J. K., Zhou, H. & Tilton, R. D. Liposome rupture and contents release over coplanar microelectrode arrays. *J. Colloid Interf. Sci.* **332**, 113-121 (2009).

[52] Chan, K. L., Gascoyne, P. R. C., Becker, F. F. & Pethig, R. Electrorotation of liposomes: Verification of dielectric multi-shell model for cells. *Biochim. Biophys. Acta, Lipids Lipid Metab.* **1349**, 182-196 (1997).

[53] Bakás, L., Chanturiya, A., Herlax, V. & Zimmerberg, J. Paradoxical lipid dependence of pores formed by the *Escherichia coli* α -hemolysin in planar phospholipid bilayer membranes. *Biophys. J.* **91**, 3748-3755 (2006).

[54] Stauffer, V., Stoodley, R., Agak, J. O. & Bizzotto, D. Adsorption of DOPC onto Hg from the GIS interface and from a liposomal suspension. *J. Electroanal. Chem.* **516**, 73-82 (2001).

TEXT ADDED: Supplementary Information Page 32-33: Supplementary Section 19

Detailed explanations on dielectric model and CM factor calculation of $A\beta_{42}$ assemblies. Based on ellipsoidal model and MWO theory.

“Calculation of CM factor of $A\beta_{42}$ assemblies. To achieve dielectric properties of $A\beta_{42}$ fibrils, an ellipsoidal approximation was employed to simplify the high aspect ratio of the fibrous structure into an ellipsoidal model (**Fig. S20a**)⁵⁵. Using equations (17) and (18), the $\text{Re}[f_{\text{CM}}(\omega)]$ of the ellipsoidal $A\beta_{42}$ fibrils (**Fig. S20b**) was estimated with a depolarizing factor of $A = 0.0047$. In this process, the surface conductance of the $A\beta_{42}$ fibrils was also considered by using the effective surface conductivity of a charged ellipsoidal particle when the long-axis is parallel to the E-field, which is given by⁵⁶,

$$\sigma_p = \sigma_{\text{bulk}} + \frac{3}{2R_y} K_{\text{surf}} \cosh \xi_0 I_1 \quad (25)$$

in which $I_1 = (1 - \sinh^2 \xi_0) \cdot \arcsin(1/\cosh \xi_0) + \sinh \xi_0$ and $\tanh \xi_0 = R_y/R_x$.

While CM factor of the $A\beta_{42}$ fibrils was calculated by ellipsoidal approximation, the spherical dielectric model together with the MWO theory was employed for $A\beta_{42}$ oligomers/protofibrils (**Fig. S20b**). Owing to the lack of literature, we assumed the surface conductance of $A\beta_{42}$ from those of biological macromolecules⁵⁷. Although the DEP force is much stronger in the direction parallel to the long-axes of spheroids, the captured $A\beta_{42}$ fibrils are observed to be aligned along the VNE edge. This is attributed to their greater structural flexibility (**Fig. S21** and **Main Figs. 5f** and **5g**) compared with stiff yeast cells (**Fig. S14**).”

FIGURE ADDED: Supplementary Fig. S20. Dielectric model and CM factor of $A\beta_{42}$ assemblies.

“Fig. S20 Dielectric model and calculated CM factor of $A\beta_{42}$ assemblies. a, b Schematic illustration of the dielectric model of $A\beta_{42}$ fibrils based on the spheroidal dielectric model (a) and the calculated $\text{Re}[f_{\text{CM}}(\omega)]$ of different types of $A\beta_{42}$ assemblies; fibril (red) and oligomer/protofibril (blue) as a function of frequency (b). Note that the f_r of the $A\beta_{42}$ fibril and the oligomer/protofibril were estimated to be 4.9 MHz and 214 MHz, respectively.”

TABLE ADDED: Supplementary Table S6. Parameters for calculation of CM factor of $A\beta_{42}$ assemblies.

REFERENCE ADDED: Supplementary references [55-59]

[55] Green, N. G. & Jones, T. B. Numerical determination of the effective moments of non-spherical particles. *J. Phys. D: Appl. Phys.* **40**, 78-85 (2006).

[56] Saville, D. A., Bellini, T., Degiorgio, V. & Mantegazza, F. An extended Maxwell–Wagner theory for the electric birefringence of charged colloids. *J. Chem. Phys.* **113**, 6974-6983 (2000).

[57] Clarke, R. W., Piper, J. D., Ying, L. & Klenerman, D. Surface conductivity of biological macromolecules measured by nanopipette dielectrophoresis. *Phys. Rev. Lett.* **98**, 198102 (2007).

[58] Al-Ahdal, S. A. *et al.* Dielectrophoresis of amyloid-beta proteins as a microfluidic template for alzheimer's research. *Int. J. Mol. Sci.* **20**, 3595 (2019).

[59] Ing, N. L., El-Naggar, M. Y. & Hochbaum, A. I. Going the distance: Long-range conductivity in protein and peptide bioelectronic materials. *J. Phys. Chem. B* **122**, 10403-10423 (2018).

TEXT ADDED: Main manuscript Page 7

“This implies that NPs with high surface area to volume ratio experience higher K_{surf} ; thus, surface conductance become dominant in the case of smaller NPs. Because MWO theory works at low conductive environments^{49,50}, the effective dielectrophoretic properties of PS NPs, including CM factor, were represented using the MWO model (Supplementary Section 10).”

REFERENCE ADDED: Main references [49-50]

[49] Nakano, A. & Ros, A. Protein dielectrophoresis: Advances, challenges, and applications. *Electrophoresis* **34**, 1085-1096 (2013).

[50] Hughes, M. P. *Nanoelectromechanics in Engineering and Biology*. (CRC Press, Boca Raton, 2003).

TEXT REVISED: Main manuscript Page 9

Comments on dielectric model of yeast cells and *B. subtilis*.

“As an example of micron-scale bioparticles, the possible capturing ranges of yeast cells and *Bacillus subtilis* spores were theoretically evaluated using spherical- and elongated multi-shell dielectric models using MWO theory, and the trapping behaviours were experimentally confirmed under low voltage amplitudes of $V_{\text{pp}} = 0.5$ and 0.8 V, respectively (Supplementary Sections 14 and 15).”

TEXT ADDED: Main manuscript Page 9

Comments on dielectric model of PS nanoparticles.

“According to the calculation of the CM factor based on the MWO model, the values of f_{χ} for each NP were calculated as 12, 18, and 37 MHz, respectively (Supplementary Section 10). From this analysis, a frequency of 10 kHz was selected to achieve ACEO dominance, and 100 kHz guaranteed DEP dominance.”

TEXT ADDED: Main manuscript Page 10-11

Comments on dielectric model of SUVs.

“To predict the frequency dependent dielectric behaviour of SUVs, the CM factor was calculated by utilizing the walled cell model designed by Jones (Supplementary Section 17), followed by experimental corroborations.”

TEXT ADDED: Main manuscript Page 11

Comments on dielectric model of A β ₄₂ assemblies.

“Prior to the experiments, the CM factors of A β ₄₂ peptides were calculated by using dielectric models depending on molecular shapes and sizes (Supplementary Section 19).”

1.4.2. *“Possible local surface charges should be commented.”*

The authors explained and replied to this concern in #1.4.1 using MWO theory. The physics of local surface charges and its effect on dielectric behavior of nanoparticles are now revised and newly included in the main manuscript.

TEXT REVISED AND ADDED: Main manuscript Page 7

Detailed explanations on effects of local surface charge on dielectric properties of nanoparticles are addressed in the revised version using MWO theory.

“The direction of F_{DEP} can also be controlled by changing the frequency because F_{DEP} highly depends on the polarizing behaviour of particles. For example, particles move along the increasing field gradient at the condition of pDEP where $\text{Re}[f_{\text{CM}}(\omega)] > 0$, whereas repulsion of particles from the region of the highest field gradient is referred to as nDEP where $\text{Re}[f_{\text{CM}}(\omega)] < 0$ ³⁴. In particular, for charged particles with sizes comparable to their EDL thickness, ionic currents and their convection within the EDL play an important role in polarization and formation of induced dipole⁴⁷. To evaluate the dielectric response of a suspended particle in the presence of an applied E-field, the Maxwell–Wagner–O’Konski (MWO) model was adopted to consider both local surface charges and the resultant surrounding EDL. In this model of MWO dielectric dispersion, the effective conductivity of particles (σ_p) with a radius of R can be described as $\sigma_p = \sigma_{\text{bulk}} + 2K_{\text{surf}}/R$, which is a combination of the bulk conductivity (σ_{bulk}) and surface conductance (K_{surf}) of particles⁴⁸. This implies that NPs with high surface area to volume ratio experience higher K_{surf} ; thus, surface conductance become dominant in the case of smaller NPs. Because MWO theory works at low conductive environments^{49,50}, the effective dielectrophoretic properties of PS NPs, including CM factor, were represented using the MWO model (Supplementary Section 10).”

REFERENCE ADDED: Main references [47-48]

[47] Green, N. G. & Morgan, H. Dielectrophoresis of submicrometer latex spheres. 1. Experimental results. *J. Phys. Chem. B* **103**, 41-50 (1999).

[48] O’Konski, C. T. Electric properties of macromolecules. V. Theory of ionic polarization in polyelectrolytes. *J. Phys. Chem.* **64**, 605-619 (1960).

1.5. Again, related to the points previously listed, no trapping scenario is proposed. Most of the time, trapping results from the balance between two forces, but could also result from the localisation of the particle at a minimum of potential related to one single particular force (as it does for electrostatic trapping). A more thorough and rigorous analysis of the physics would help regarding this.

The authors strongly agree with reviewer's comment that the particle localization within the electrokinetic forces could be expressed in terms of minimum of potential energy (U). As an effort to explain localization of particle on particular region of the electrodes, the authors have conducted a calculation of the potential energy landscapes along the electrode. In this process, electrostatic potential energies under particle capturing conditions ($f = 10$ & 100 kHz) are calculated from inverse gradient of F_{DEP} and F_{ACEO} , using relation of $F = -\nabla U$. The detailed description is newly represented in Supplementary Information and briefly mentioned at the revised main manuscript.

The potential energy along the surface of electrode was discussed and its values were calculated under the application of frequencies at $f = 10$ kHz and 100 kHz.

TEXT ADDED: Supplementary section 12

“Particles suspended in an aqueous environment move and migrate to positions at which the potential energy is minimum. As two dominant forces affect the local position of the suspended particles on the VNE surface, the total potential energy profile along the surface, $U(x)$, can be given as $U(x) = U_{\text{DEP}}(x) + U_{\text{ACEO}}(x)$, where U_{DEP} and U_{ACEO} indicate that the potential energies arise from DEP- and ACEO driving forces. Note that two dominant potential energies are achieved from the inverse gradient of F_{DEP} and F_{ACEO} using the equations $U_{\text{DEP}} = -\pi\epsilon_m R^3 \text{Re}[f_{\text{CM}}(\omega)]E^2$ and $F_{\text{ACEO}} = -\nabla U_{\text{ACEO}}$, respectively^{23,28}. Shown in **Fig. S10**, the minimum value of total potential energy $U(x)$ is calculated at the centre of the VNE at $f = 10$ kHz (**Fig. S10a**), in contrast to the edges of the VNE at $f = 100$ kHz (**Fig. S10b**), respectively. This shows great consistency with the experimental demonstration shown in **Main Fig. 3**.”

REFERENCE ADDED: Supplementary references [28]

[28] Pethig, R. R. *Dielectrophoresis: Theory, Methodology and Biological Applications*. (John Wiley & Sons, Hoboken, 2017).

FIGURE ADDED: Supplementary Fig. S10 (newly added).

“**Fig. S10 Potential energy landscape on VNE surface. a, b** Calculated $U_{\text{DEP}}(x)$ (red), $U_{\text{ACEO}}(x)$ (blue), and $U(x)$ (black) along the bottom electrode surface of VNE ($d = 100$ nm) under $V_{\text{pp}} = 2.5$ V, $f = 10$ kHz (**a**), and 100 kHz (**b**).”

TEXT REVISED: Main Manuscript Page 8

“Then, because the ACEO-induced slip velocity (pushing toward centre) is found to be stronger than the DEP force (attracting toward edges) at the collecting surface of the VNE, effective surface streams push particles toward the stagnation point where the potential energy is the lowest (Supplementary Section 12)⁵¹. In contrast, at 100 kHz (Fig. 3h) and 1 MHz (Fig. 3i) in regime II, the ring-shaped particle assembly was observed at the edge of nanogap, where the value of ∇E^2 is the largest and the potential energy is the lowest (Supplementary Section 12).”

REFERENCE ADDED: Main references [51]

[51] Liu, S.-J., Wei, H.-H., Hwang, S.-H. & Chang, H.-C. Dynamic particle trapping, release, and sorting by microvortices on a substrate. *Phys. Rev. E* **82**, 026308 (2010).

1.6. Brownian motion and molecular diffusion are kept silent throughout the paper. I agree that they are probably negligible in most cases, but quantitative comments must be made on this specific point, especially if the particles' Stokes numbers are small.

The authors agree with reviewer #1 that quantitative analysis of Brownian motion and diffusion on our proposed VNE should be made for analyzing particle dynamics in aqueous environment. In this regard, dimensionless analysis on Péclet number for mass transport is carried out. Péclet number is a non-dimensional number, which is defined as a transport ratio advection- to diffusion rate. Therefore, under simultaneous interaction of external driving forces and Brownian diffusion, the study of particle transportation in a continuum can be estimated using Péclet number for mass transfer. The authors newly added comments for this issue in the main manuscript and supplementary information.

Discussion on Péclet number, its calculation, and effects of Brownian diffusion are commented in detail in Supplementary Information and briefly mentioned in Main Manuscript.

TEXT ADDED: Supplementary Information Page 10

Calculation of Péclet number (newly added).

“Péclet number of particles for estimating effect of Brownian motion and diffusion. The standard description of the Langevin equation on particles includes external driving forces and random Brownian force ($\zeta(t)$). When the advective and diffusive transportation of suspended particles occur simultaneously, the motion of moving particles in the fluid can be presumed using Péclet number for mass transfer (Pe), defined as the ratio of transport rate of advection to diffusion. A system with $Pe \gg 1$ governs the movement of suspended particles depending on the external driving forces (ACEO or DEP for our case), while the motions of particles are dominated by the Brownian diffusion for $Pe \ll 1$. Utilizing the mass diffusion coefficient (D) from Stokes–Einstein relationship where $D = k_B T / 6\pi\eta R$ (k_B for Boltzmann coefficient), Pe for mass transfer is given by

$$Pe = Lu/D \quad (10)$$

where L and u represent characteristic length of the system (in the order of $\sim 10 \mu\text{m}$) and fluid velocity, respectively. Since the simulated velocities were typically in the order of $\sim 10^{-4}$ m/s (Section 4 and Main Fig. 3), Pe of PS particles was calculated to be $Pe \gg 1$ for most of the cases; $Pe \approx 2 \times 10^3$, 4×10^2 , 2×10^2 , and 1×10^2 for PS particles with its diameters of 1 μm , 200 nm, 100 nm, and 50 nm, respectively. Therefore, we concluded that the effects of diffusion are negligible relative to the driving force, thereby the stochastic term $\zeta(t)$ can be neglected from the Langevin equation²³.”

REFERENCE ADDED: Supplementary references [23]

[23] Juniper, M. P. N., Straube, A. V., Aarts, D. G. A. L. & Dullens, R. P. A. Colloidal particles driven across periodic optical-potential-energy landscapes. *Phys. Rev. E* **93**, 012608 (2016).

TEXT ADDED: Main Manuscript Page 4

“To evaluate the competitive forces and resultant dynamics of a spherical particle in an aqueous environment, the Langevin equation of particle velocity (\mathbf{u}_p) was employed³⁵:

$$m_p \frac{d\mathbf{u}_p}{dt} = \mathbf{F}_{\text{DEP}} + \mathbf{F}_{\text{ACEO}} + \mathbf{F}_{\text{ETF}} + \mathbf{F}_{\text{grav}} + \mathbf{F}_{\text{buoy}} + \zeta(t) + \sum \mathbf{F}_{i,j} \quad (3)$$

considering DEP force (\mathbf{F}_{DEP}), Stokes drag force induced by ACEO flow (\mathbf{F}_{ACEO}) and electrothermal flow (\mathbf{F}_{ETF}), gravitational force (\mathbf{F}_{grav}), buoyant force (\mathbf{F}_{buoy}), random Brownian force ($\zeta(t)$), and interaction between neighbouring particles ($\mathbf{F}_{i,j}$; interaction force acting on the i -th particle owing to the j -th particles). Because Stokes drag forces are defined as $\mathbf{F} = -6\pi\eta R(\mathbf{u}_p - \mathbf{u}_m)$, where \mathbf{u}_p and \mathbf{u}_m are the velocities of particles and fluidic flows, respectively, \mathbf{F}_{ACEO} and \mathbf{F}_{ETF} exerting on a single particle can be evaluated by inserting \mathbf{u}_{ACEO} and \mathbf{u}_{ETF} into \mathbf{u}_m , respectively³⁶. Considering the simulations (\mathbf{F}_{ETF} , \mathbf{F}_{DEP} , and \mathbf{F}_{ACEO}), calculations (\mathbf{F}_{grav} , \mathbf{F}_{buoy} , and $\zeta(t)$), and low concentration (10 ppm) condition ($\mathbf{F}_{i,j}$) on the VNE having pattern size (L) and periodicity (P) of $L = 10 \mu\text{m}$ and $P = 30 \mu\text{m}$ (**Fig. 1b**), \mathbf{F}_{DEP} and \mathbf{F}_{ACEO} dominantly determine the movement of 1- μm -diameter polystyrene (PS) particles, while the others are negligible (Supplementary Sections 1-5).”

1.7. Under AC fields, particles will acquire an electric dipole moment (especially at high frequency, where Debye layer cannot screen the charges), which will trigger dipole interactions and subsequent interaction forces (attractive or repulsive depending on the relative positions of each particles, but most of the time attractive). Again, no comment is made about this potentially cluster generating process.

As reviewer #1 noted, induced dipole moments of particles trigger interactive forces. Although their interactions are strong between adjacent particles, they become negligible for distant suspended particles with relatively low concentration. In this end, inter-particulate forces are considered negligible in Langevin equation, and the simulations were carried out without them. However, since inter-particulate attracting interaction occurs in-between highly populated particles, comments should be made on the experimental situations of particle trapping. The interactive clustering is commented in the main manuscript, but quantitative analysis is not conducted since it is out of the scope of the article,

Inter-particulate interactions are newly commented in Langevin equation and experimental part of the main manuscript.

TEXT ADDED: Main manuscript Page 4

“To evaluate the competitive forces and resultant dynamics of a spherical particle in an aqueous environment, the Langevin equation of particle velocity (\mathbf{u}_p) was employed³⁵.

$$m_p \frac{d\mathbf{u}_p}{dt} = \mathbf{F}_{\text{DEP}} + \mathbf{F}_{\text{ACEO}} + \mathbf{F}_{\text{ETF}} + \mathbf{F}_{\text{grav}} + \mathbf{F}_{\text{buoy}} + \zeta(t) + \sum \mathbf{F}_{i,j} \quad (3)$$

considering DEP force (\mathbf{F}_{DEP}), Stokes drag force induced by ACEO flow (\mathbf{F}_{ACEO}) and electrothermal flow (\mathbf{F}_{ETF}), gravitational force (\mathbf{F}_{grav}), buoyant force (\mathbf{F}_{buoy}), random Brownian force ($\zeta(t)$), and interaction between neighbouring particles ($\mathbf{F}_{i,j}$; interaction force acting on the i -th particle owing to the j -th particles). Because Stokes drag forces are defined as $\mathbf{F} = -6\pi\eta R(\mathbf{u}_p - \mathbf{u}_m)$, where \mathbf{u}_p and \mathbf{u}_m are the velocities of particles and fluidic flows, respectively, \mathbf{F}_{ACEO} and \mathbf{F}_{ETF} exerting on a single particle can be evaluated by inserting \mathbf{u}_{ACEO} and \mathbf{u}_{ETF} into \mathbf{u}_m , respectively³⁶. Considering the simulations (\mathbf{F}_{ETF} , \mathbf{F}_{DEP} , and \mathbf{F}_{ACEO}), calculations (\mathbf{F}_{grav} , \mathbf{F}_{buoy} , and $\zeta(t)$), and low concentration (10 ppm) condition ($\mathbf{F}_{i,j}$) on the VNE having pattern size (L) and periodicity (P) of $L = 10 \mu\text{m}$ and $P = 30 \mu\text{m}$ (**Fig. 1b**), \mathbf{F}_{DEP} and \mathbf{F}_{ACEO} dominantly determine the movement of 1- μm -diameter polystyrene (PS) particles, while the others are negligible (Supplementary Sections 1-5).”

TEXT ADDED: Page 8

“Note that the interparticle force ($\sum \mathbf{F}_{i,j}$) component is negligible when suspended particles are distant from each other. However, when they are trapped and come together (**Figs. 3g** and **3h**), particle–particle interactions improve the trapping efficiency of the dielectrophoresis devices by the induced dipole moments of particles^{35,52,53}.”

REFERENCE ADDED: Main references [52-53]

[52] Israelachvili, J. N. *Intermolecular and Surface Forces*. (Academic Press, Burlington, 2011).

[53] Lee, D.-H., Yu, C., Papazoglou, E., Farouk, B. & Noh, H. M. Dielectrophoretic particle–particle interaction under AC electrohydrodynamic flow conditions. *Electrophoresis* **32**, 2298-2306 (2011).

1.8. There are big gaps in the literature. Articles by A. Ajdari, Castellanos and Squires, among others, should be mentioned.

The authors would like to thank reviewer #1 for providing suggestions to improve depth of our work. After reading the suggested literatures, the author added and cited related references in both main manuscript (28 added & 12 removed) and Supplementary information (34 added & 3 removed).

References are added including works of A. Ajdari, Castellanos, Squires, and the others.

REFERENCE ADDED/REMOVED/MOVED: 28 main references are added and 12 main references are removed to improve depth of our work. Besides, several references are moved into more appropriate positions and reference numbers are rearranged accordingly.

REFERENCE ADDED: Main references [6], [10], [12], [16], [19], [23], [26], [36], [37], [39-45], [47-53], [61] and [64-67]

[6] Adato, R. & Altug, H. *In-situ* ultra-sensitive infrared absorption spectroscopy of biomolecule interactions in real time with plasmonic nanoantennas. *Nat. Commun.* **4**, 2154 (2013).

[10] Stone, H. A., Stroock, A. D. & Ajdari, A. Engineering flows in small devices: Microfluidics toward a lab-on-a-chip. *Annu. Rev. Fluid Mech.* **36**, 381-411 (2004).

[12] Reimann, P. Brownian motors: Noisy transport far from equilibrium. *Phys. Rep.* **361**, 57-265 (2002).

[16] Squires, T. M. & Quake, S. R. Microfluidics: Fluid physics at the nanoliter scale. *Rev. Mod. Phys.* **77**, 977-1026 (2005).

[19] Junno, T., Deppert, K., Montelius, L. & Samuelson, L. Controlled manipulation of nanoparticles with an atomic force microscope. *Appl. Phys. Lett.* **66**, 3627-3629 (1995).

[23] Squires, T. M. Induced-charge electrokinetics: Fundamental challenges and opportunities. *Lab Chip* **9**, 2477-2483 (2009).

[26] Khoshmanesh, K., Nahavandi, S., Baratchi, S., Mitchell, A. & Kalantar-zadeh, K. Dielectrophoretic platforms for bio-microfluidic systems. *Biosens. Bioelectron.* **26**, 1800-1814 (2011).

[61] Chen, X. *et al.* Atomic layer lithography of wafer-scale nanogap arrays for extreme confinement of electromagnetic waves. *Nat. Commun.* **4**, 2361 (2013).

[64] Ding, S.-Y. *et al.* Nanostructure-based plasmon-enhanced Raman spectroscopy for surface analysis of materials. *Nat. Rev. Mater.* **1**, 16021 (2016).

[65] Adato, R., Aksu, S. & Altug, H. Engineering mid-infrared nanoantennas for surface enhanced infrared absorption spectroscopy. *Mater. Today* **18**, 436-446 (2015).

[66] Homola, J. Surface plasmon resonance sensors for detection of chemical and biological species. *Chem. Rev.* **108**, 462-493 (2008).

[67] Seo, M. & Park, H.-R. Terahertz biochemical molecule-specific sensors. *Adv. Opt. Mater.* **8**, 1900662 (2020).

Details of references including [36], [37], [39-45], [47-53] are already described above.

REFERENCE REMOVED: Main references

[6], [14], [15], [19-21], [27], [38], and [48-51] in previous version

REFERENCE MOVED: Main references

[22] Pohl, H. A. *Dielectrophoresis: The Behavior of Neutral Matter in Nonuniform Electric Fields.* (Cambridge University Press, Cambridge, 1978). ← moved from [11] of previous version

[24] Nadappuram, B. P. *et al.* Nanoscale tweezers for single-cell biopsies. *Nat. Nanotechnol.* **14**, 80-88 (2019). ← moved from [28] of previous version

[31] Pohl, H. A. The motion and precipitation of suspensoids in divergent electric fields. *J. Appl. Phys.* **22**, 869-871 (1951). ← moved from [24] of previous version

[32] Ramos, A., Morgan, H., Green, N. G. & Castellanos, A. Ac electrokinetics: A review of forces in microelectrode structures. *J. Phys. D: Appl. Phys.* **31**, 2338-2353 (1998). ← moved from [25] of previous version

[38] Green, N. G., Ramos, A., González, A., Morgan, H. & Castellanos, A. Fluid flow induced by nonuniform ac electric fields in electrolytes on microelectrodes. III. Observation of streamlines and numerical simulation. *Phys. Rev. E* **66**, 026305 (2002). ← moved from [37] of previous version

[46] Oh, J., Hart, R., Capurro, J. & Noh, H. M. Comprehensive analysis of particle motion under non-uniform AC electric fields in a microchannel. *Lab Chip* **9**, 62-78 (2009). ← moved from [33] of previous version

REFERENCE ADDED: Several supplementary references are added (34 references including [2], [5], [22-24], [28-36], [39-43], and [45-59]) and removed (3 references including [2], [5], and [26] in previous version) to Supplementary Information and reference numbers are rearranged accordingly. Details of references are already described above.

REFERENCE ADDED: Supplementary references

[2] Henningses, J., Huenges, E. & Burkhardt, H. In situ thermal conductivity of gas-hydrate-bearing sediments of the Mallik 5L-38 well. *J. Geophys. Res.: Solid Earth* **110** (2005).

[5] Kwon, Y.-W., Lee, C. H., Choi, D.-H. & Jin, J.-I. Materials science of DNA. *J. Mater. Chem.* **19**, 1353-1380 (2009).

Details of references including [22-24], [28-36], [39-43], and [45-59] are already described above.

REFERENCE REMOVED: Supplementary references

[2], [5], [26] in previous version

2. POINT-BY-POINT RESPONSE TO COMMENTS FROM REVIEWER #2

2.1. The manuscript in its present form needs to be proofread (I have included a highlighted copy of the manuscript and Supplementary Information), to improve its layout and of the Supplementary Information. There is no Introduction/Background section header, and the figures of the Supplementary Information are either miss-referenced and positioned in an order that makes the reading hard going back and forth.

The authors sincerely apologize for our mistakes and also appreciate reviewer #2 not only to provide constructive comments but also one more chance to improve quality of our work. The authors also thank reviewer #2's kindness in correcting the typos in the previous version despite lack of its readability. Now in the revised version, the authors have rearranged entire order of contents and corrected those mistakes for improvement of legibility.

1. *Introduction section header was added to the revised main manuscript.*
2. *Typos pointed out by reviewer #2 were revised (marked as blue highlight). Also, to improve layout of Supplementary Information, descriptions and figures in Supplementary Information were grouped in order according to the subjects.*
3. *To improve readability, the layout of Supplementary Information was rearranged, accompanied with significant amount of newly added data and related contents that the reviewers had requested to answer.*

Supplementary Information Page 2: Supplementary Section 1

TEXT MOVED/REVISED: Model for numerical simulations (Page 5 in previous version) → As a description in Supplementary Section 1 (revised).

TABEL MOVED/REVISED: Table S1 (Page 5 in previous version) → As a Supplementary Table S1 (revised).

FIGURE ADDED: Supplementary Fig. S1 (newly added).

Supplementary Information Page 3-5: Supplementary Section 2

TEXT MOVED/REVISED: Simulations of dielectrophoresis (DEP) & Simulations of AC electro-osmosis (ACEO) (Pages 5-7 in previous version) → As a description in Supplementary Section 2 (re-ordered and revised).

FIGURE ADDED/REVISED: Fig. S1 (Page 10 in previous version) → As a Supplementary Fig. S2 (revised).

Supplementary Information Page 6-7: Supplementary Section 3

TEXT MOVED/REVISED: Simulations of Joule heating & the electrothermal effect (Page 7-8 in previous version) → As a description in Supplementary Section 3 (re-ordered and revised).

FIGURE ADDED/REVISED: Fig. S1 (Page 10 in previous version) → As a Supplementary Fig. S3 (revised).

Supplementary Information Page 8-9: Supplementary Section 4

TEXT MOVED/REVISED: Relationship between fluid velocity into force (Page 9 in previous version) → As a description in Supplementary Section 4 (re-ordered and revised).

FIGURE ADDED/REVISED: Figs. S2 (Page 11) and S5 (Page 13) in previous version → As a Supplementary Fig. S4 (revised). Re-simulated F_{DEP} and F_{ACEO} under changed conditions are included, and replaced F_{total} into F_{ETF} under different frequencies.

Supplementary Information Page 10: Supplementary Section 5

TEXT MOVED/REVISED: Calculation of gravitational forces (Page 8-9 in previous version) → As a description in Supplementary Section 5 (revised).

TEXT ADDED: Calculation of Brownian motion (newly added).

Supplementary Information Page 11: Supplementary Section 6

TEXT ADDED: Description on Stokes number (newly added).

Supplementary Information Page 12: Supplementary Section 7

TEXT MOVED: Several sentences in figure legend of Fig. S3 (Page 11-12) → As a description in Supplementary Section 7 (re-ordered and revised).

FIGURE MOVED: Fig. S3 (Page 11-12) → Supplementary Figure S5 (revised).

Supplementary Information Page 13: Supplementary Section 8.

FIGURE ADDED: Supplementary Figs. S6 and S7 (newly added)

Supplementary Information Page 14-15: Supplementary Section 9

TEXT MOVED: Fabrication process of vertical nanogap electrode (VNE) (Page 2 in previous version) → As a description in Supplementary Section 9

FIGURE MOVED/REVISED: Fig. S4 (Page 12 in previous version) → Supplementary Fig. S8

Supplementary Information Page 16-17: Supplementary Section 10

TEXT MOVED/REVISED: Preparation of PS micro/nanoparticles & calculation of CM factor of PS micro/nanoparticles. (Page 15 in previous version) → As a description in Supplementary Section 10 (revised)

TABLE ADDED: Table S2 (newly added)

FIGURE MOVED/REVISED: Figs. S6 (Page 14 in previous version) → As a Supplementary Fig. S9 (revised)

Supplementary Information Page 18: Supplementary Section 11

TEXT MOVED/REVISED: Voltage application and data analysis & SEM observation (Page 4 in previous version) → As a description in Supplementary Section 11 (revised)

Supplementary Information Page 19: Supplementary Section 12

TEXT ADDED: Description on spatial potential energy (newly added).

FIGURE ADDED: Supplementary Fig. S10 (newly added).

Supplementary Information Page 20-21: Supplementary Section 13

FIGURE MOVED/REVISED: Figs. S7 (Page 16) and S8 (Page 17) in previous version → As a Supplementary Figs. S11 and S12, respectively (revised).

Supplementary Information Page 22-24: Supplementary Section 14

TEXT ADDED: Description on yeast preparation & CM factor of yeast (newly added).

TABLE ADDED: Supplementary Table S3 (newly added).

FIGURE ADDED: Supplementary Figs. S13 and S14 (newly added).

Supplementary Information Page 25-26: Supplementary Section 15

TEXT MOVED: Preparation of *Bacillus subtilis* spores (Page 3 in previous version) → As a description in Supplementary Section 15.

TEXT ADDED: Description on CM factor of *B. subtilis* (newly added).

TABLE ADDED: Supplementary Table S4 (newly added).

FIGURE ADDED: Supplementary Figs. S15 (newly added).

FIGURE MOVED/REVISED: Fig. S9 (Page 17 in previous version) → As a Supplementary Fig. S16 (revised).

Supplementary Information Page 27: Supplementary Section 16

TEXT MOVED/REVISED: Preparation of PS micro/nanoparticles (Page 2 in previous version) → As a description in Supplementary Section 16 (revised).

FIGURE MOVED/REVISED: Fig. S10 (Page 18 in previous version) → As a Supplementary Fig. S17 (revised).

Supplementary Information Page 28-29: Supplementary Section 17

TEXT MOVED/REVISED: Preparation of nanovesicles (Page 3 in previous version) → As a description in Supplementary Section 17 (revised).

TEXT ADDED: Description on CM factor of SUV (newly added).

FIGURE ADDED: Supplementary Figs. S18 (newly added).

TABLE ADDED: Supplementary Table S5 (newly added).

Supplementary Information Page 30-31: Supplementary Section 18

TEXT ADDED: Description on large-area performance analysis (newly added).

FIGURE ADDED: Supplementary Figs. S19 (newly added).

Supplementary Information Page 32-33: Supplementary Section 19

TEXT MOVED: Preparation of A β ₄₂ assemblies (Page 3 in previous version) → As a description in Supplementary Section 19.

TEXT ADDED: Description on CM factor A β ₄₂ assemblies (newly added).

FIGURE ADDED: Supplementary Figs. S20 (newly added).

TABLE ADDED: Supplementary Table S6 (newly added).

FIGURE MOVED/REVISED: Fig. S12 (Page 19 in previous version) → As a Supplementary Figs. S21 (revised).

Supplementary Information Page 34: Supplementary Section 20

TEXT MOVED: CD spectrometer and DLS measurement (Page 4 in previous version) → As a description in Supplementary Section 20.

FIGURE MOVED/REVISED: Fig. S13 (Page 19 in previous version) → As a Supplementary Figs. S22 (revised).

Supplementary Information Page 35: Supplementary Movies

TEXT MOVED/REVISED: Legends in Supplementary Movies (Page 19 in previous version) → As a Legends in Supplementary Movies (revised).

Supplementary Information Page 35-39: Supplementary References

TEXT REVISED: References → Supplementary references.

REFERENCES ADDED: Former references are rearranged. 34 number of references are added and 3 reference is removed. The references numbers are changed accordingly.

2.2. The authors should include repetitions of their experiments with the different (or a subset of) NPs and provide a standard deviation or error for the normalized intensity or threshold potential. Additionally, I would be very interested in seeing the device robustness in terms of the fabrication process (i.e. evaluate the large area performance of the device by measuring FL intensity at different regions of the VNEs array).

The authors sincerely appreciate reviewer #2 on mentioning the device’s reliability for evenly capturing over a large area. Since our VNE emphasizes its large-area applicability, the authors agree that the previous manuscript was insufficient to emphasize large-area performance consistency in a view point of FL uniformity. In this regard, the authors have re-analyzed the VNE performances from multiple-unit cells and provided statistical results with their intermediate process of analysis. The newly added data obtained from large-area analysis confirm a uniformity as well as massive capturing performance of the proposed VNE.

2.2.1. “The authors should include repetitions of their experiments with the different (or a subset of) NPs and provide a standard deviation or error for the normalized intensity or threshold potential.”

For trapping PS nanoparticles, time-lapse variation of normalized FL intensities and threshold voltages were analyzed on multiple unit cells of $N = 49$ and presented with their statistical results. Also for SUVs, same procedure was carried out on multiple unit cells of $N = 81$. The normalized FL intensity on a single unit cell was replaced into that of averaged values on multiple unit cells. The standard deviations of threshold voltages were calculated and given as the error bars in inset graphs.

Main Figures:

FIGURE REVISED: Main Figs. 4e-4h in previous version → Main Figs. 4e & inset (revised).

- Time-lapse FL intensities of PS NPs on an arbitrary single unit cell → Averaged time-lapse FL intensities of PS NPs on 49 (7 x 7) unit cells.
- Threshold voltage of PS NPs on an arbitrary single unit cell → Averaged threshold voltages of PS NPs on 49 unit cells (inset) & their standard deviations (given as error bars in inset).

FIGURE LEGEND REVISED: Fig. 4e were revised accordingly.

“e Averaged FL intensities of captured PS NPs on 49 unit cells as a result of increasing V_{pp} ($f = 100$ kHz) from 0 to 3 V by 0.1 V for every 30 s; averaged trapping thresholds of 0.33, 0.72, and 1.24 V for 200, 100, and 50 nm, respectively (inset).”

FIGURE REVISED: Main Fig. 5a → Main Figs. 5a & inset (revised).

- Time-lapse FL intensities of SUVs on an arbitrary single unit cell → Averaged time-lapse FL intensities of SUVs on 81 (9 x 9) unit cells.
- Threshold voltage of SUVs on an arbitrary single unit cell → Averaged threshold voltages of SUVs on 81 unit cells (inset) & their standard deviations (given as errors bar in inset).

FIGURE LEGEND REVISED: Fig. 5 legend were revised accordingly.

“**Fig. 5 Trapping of nanoparticulate biomaterials on the VNE with large-area applicability and design flexibility.** **a** Averaged FL intensities and corresponding FL micrographs of captured 50-nm SUVs on 81 unit cells as a result of increasing V_{pp} ($f = 100$ kHz) from 0 to 5 V by 0.1 V every 10 s; averaged trapping threshold of 1.76 V (inset). Scale bars, 10 μm .”

TEXT REVISED/ADDED: Main manuscript Page 9-10.

“In contrast, at 100 kHz, only 200 nm PS particles started to be captured at the nanogap from the averaged trapping threshold (V_{th}) of $V_{th} = 0.33$ V (Figs. 4b and 4e). As the AC voltage increased, the 100 nm particles began to be captured at $V_{th} = 0.72$ V while no 50 nm particles were monitored (Figs. 4c and 4e). Finally, the trapping of the 50 nm particles was observed for $V_{th} = 1.24$ V (Figs. 4d and 4e) while other larger particles became highly packed (see the SEM image of Supplementary Section 16). In this process, the time-lapse normalized FL intensities were analysed and averaged for a VNE unit cell number (N) of $N = 49$, and the standard deviations of V_{th} were calculated to be 0.11, 0.12, and 0.22 V for PS NPs of 200, 100, and 50 nm, respectively (Fig. 4e). Note that a glitch occurred at the moment when electrical signals changed owing to equipment limitations.”

TEXT REVISED/ADDED: Main manuscript Page 10.

“As the applied V_{pp} increased by 0.1 V every 10 s at a fixed frequency of 100 kHz, the SUVs began to be captured from $V_{th} = 1.76$ V ($N = 81$, standard deviation of 0.23 V), and the amount captured increased accordingly with time and voltage amplitude (Fig. 5a).”

2.2.2. “Additionally, I would be very interested in seeing the device robustness in terms of the fabrication process (i.e. evaluate the large area performance of the device by measuring FL intensity at different regions of the VNEs array).”

The authors have analyzed normalized FL intensity of the captured SUVs from multiple-unit cell to confirm uniformity and massive capturing performance of the proposed VNE. Large-area FL micrographs are included in the main manuscript, and their statistical analyses are explained in detail in Supplementary Information.

Main Figures:

FIGURE MOVED AND LEGEND REVISED: Supplementary Fig. S11 (Page 18 in previous version) → Main Figs. 5b and 5c (revised).

“**b, c** FL micrographs of 50-nm SUVs trapped at centre (**b**) and edge (**c**) of unit cells over large-area under $f = 10$ kHz and 100 kHz, respectively ($V_{pp} = 2$ V). Scale bars, 200 μm .”

TEXT REVISED: Main Manuscript Page 10:

“Moreover, ACEO- and DEP-induced trapping of 50 nm SUVs to the centre and edge of the VNEs were achieved at 10 kHz and 100 kHz, respectively, under a low voltage of 2 V over the millimetre-scale large-area (**Figs. 5b and 5c**). From an analysis of the FL intensities on 400 unit cells, the variations of normalized FL intensities on each cell demonstrated uniformity and device robustness over a large-area (Supplementary Section 18).”

TEXT ADDED: Supplementary Information Page 30-31: Supplementary Section 18

- Description on large-area analyzing process (newly added). Analyzing process of large-area FL intensities were explained in detail in Supplementary Section 18.

“Section 18: Large-area FL analysis for high-throughput and uniform trapping on VNE.”

To evaluate the large-area performance, FL microscopic images of captured SUVs on an array of $M \times N$ VNE unit cells were analyzed through a numerical analysis (Matlab, Mathworks). The FL images were prepared in the presence of applied AC voltages of $f = 10$ and 100 kHz ($V_{pp} = 2$ V shown in **Main Figs. 5b and 5c**) showing trapping of 50 nm SUV on multi-unit cells (20×20) over a large area ($0.8 \text{ mm} \times 0.8 \text{ mm}$) without any false electrode. For total pixel numbers of $MX \times NY$ (X and Y are pixel periodicities along x - and y -axis), the FL distribution of the images can be expressed as $I(x, y)$ with x and y in the ranges of $0 < x < MX$ and $0 < y < NY$. Note that x and y are integers that represent the x - and y -coordinates of the pixel positions. Then, the distribution of the FL intensity achieved from a single cell in the m -th column and n -th row can be expressed as $I_{(m,n)}(x, y) = I(x + X(m - 1), y + Y(n - 1))$ where $0 < x < X$ and $0 < y < Y$ (**Fig. S19a**; Step 1). By carrying out image-processing and numerical analysis (**Fig. S19a**; Step 2), the average FL distribution ($I_{\text{avg}}(x, y)$) was calculated from every $I_{(m,n)}(x, y)$, as follows

$$I_{\text{avg}}(x, y) = \frac{\sum_m^M \sum_n^N I_{(m,n)}(x, y)}{MN} \quad (21)$$

Subsequently, by summing the FL intensities of all pixels from $I_{\text{avg}}(x,y)$ and dividing them by the total pixel number of the unit cell, the average FL intensity of $I_{\text{avg}}(x,y)$ (I_{avg}) was calculated (Fig. S19a; Step 3). It is given by,

$$I_{\text{avg}} = \frac{\sum_x \sum_y I_{\text{avg}}(x,y)}{XY} \quad (22)$$

For minimum and maximum FL intensities of 0 and 255 (arbitrary unit), I_{avg} values were calculated to be 66 and 57 for $f = 10$ and 100 kHz, respectively (Figs. S19b and S19e). After obtaining average FL intensities of each unit cell ($I_{(m,n)}$) as follows (Fig. S19a; Step 4),

$$I_{(m,n)} = \frac{\sum_x \sum_y I_{(m,n)}(x,y)}{XY} \quad (23)$$

normalized FL intensities of each unit cell ($I_{(m,n)}^*$) were calculated by dividing each $I_{(m,n)}$ by I_{avg} (Fig. S19a; Steps 5 and 6),

$$I_{(m,n)}^* = I_{(m,n)} / I_{\text{avg}} \quad (24)$$

over 20×20 array (Figs. S19c and S19f). The standard deviations of 0.054 ($f = 10$ kHz) and 0.052 ($f = 100$ kHz) indicate that 90 % of the unit cells exhibit variations in FL intensity within ± 10 %, demonstrating the large-area uniformity of the proposed VNE array (Figs. S19d and S19g)."

FIGURE ADDED: Supplementary Fig. S19 (newly added).

“**Fig. S19 Large-area FL analysis of SUV trapping on VNE. a** Schematics of image-processing and numerical analysis of FL micrographs. **b-g** FL analysis on 400 VNE unit cells capturing 50 nm-diameter SUVs at the centre (**b-d**, $f = 10$ kHz) and edge (**e-g**, $f = 100$ kHz) under $V_{pp} = 2$ V. Averaged FL distribution with its intensity and central FL intensity profiles (white lines) (**b**, **e**), normalized FL intensities on an array of 20×20 unit cells (**c**, **f**), and their FL histogram graphs (**d**, **g**).”

2.3. The authors can also mention how they prepared the 200 um layer of PDMS (spin coating, fixed volume over fixed area) in the Supplementary Information.

The authors agree that the process of preparing thin PDMS film is insufficient, and its fabrication method is described in more details.

The manuscript of Supplementary Information was revised to provide detailed fabrication process of PDMS film.

TEXT REVISED: Supplementary Information Page 18

“After mixing a poly(dimethylsiloxane) (PDMS) elastomer and a curing agent (Sylgard 184, Dow Corning) at a ratio of 10:1 with a total weight of 3 g, the mixture was poured and evenly spread on a petri dish (size of 120 mm × 120 mm) and cured on a hotplate (80 °C for 3 h) to achieve a 200-μm-thick PDMS film.”

ADDITIONAL CHANGES:

1. The title of the manuscript is revised to emphasize the importance of our finding. A word ‘*architecture*’ is now added at the end of the previous title.
2. Typos and some English expressions were revised to increase its readability.
3. Several sentences in the main text were shortened to obey the maximum word limit of 5000, following “Brief guide for submission to *Nature Communications*”.
4. Several references in the main text were removed to obey the maximum limit of 70 references, following “Brief guide for submission to *Nature Communications*”.
5. The legends in the main text and Supplementary Information were modified.
 - a. The size of the scale bar in each figure was defined in the legend.
 - b. The references in Supplementary Information were denoted as ‘Supplementary References’.
6. **Young Tae Byun** (*Sensor System Research Center, Korea Institute of Science and Technology, Seoul 02792, Republic of Korea*) was included as a co-author owing to his contribution to the simulation and preparation of the revised manuscript.
7. Acknowledgements of **Yong-Sang Ryu** and **Seung-Yeol Lee** were revised.

“This work was supported in part by KIST intramural grants (No. 2I23820, 2V07880 and 2E30140).”

“S.-Y.L acknowledges the support by the National Research Foundation of Korea (NRF) grant funded by the Korea government Ministry of Science and ICT (No. 2017R1C1B2003585)”
8. Acknowledgements of **Minah Seo** and **Young Tae Byun** were included.

“M.S., National Research Foundation of Korea (NRF) Global Frontier Program (No. CAMM-2019M3A6B3030638)”

“Y.T.B., National Research Council of Science & Technology (NST) funded by the Korea government Ministry of Science and ICT (No. QLT-CRC-18-02-KICT)”
9. Page numbers were included in Main manuscript and Supplementary Information.
10. Several equation formats were replaced from Mathtype into normal text.
11. Texts and Figures related to simulations were revised according to the changed simulation conditions. Especially, experiments for **Figs. 3f-3j** and **supplementary video 1** were newly conducted under application of 2.5 V.
12. Data availability statement was included in Main manuscript, following “Nature research editorial policy checklist”.

“**Data availability** The data that support the findings of this study are available from the corresponding author upon reasonable request.”

REVIEWERS' COMMENTS second round:

Reviewer #1 (Remarks to the Author):

The revised version of the manuscript has been definitely improved and the objective pursued by the authors now appears more clearly to me. Considering that the article has possible interesting applications in microfluidics and bio/nanotechnologies, I reckon that the article could be suitable for publication in Nature Communication. However, I still have some minor comments that should be addressed.

1. The authors define a potential for a dissipative force or a set of dissipative forces (hydrodynamic interactions with the EO flow and DEP mainly). While I understand why they are doing this, they should mention that this is not a real potential (forces are not conservative). They could use the term 'pseudo-potential' for instance to stress that this potential is not defined in a rigorous way mechanically speaking.
2. The Langevin equation is written in the inertial full form. The authors should say early in the derivation (actually as soon as they write the equation) that they are aware that the ballistic regime is of short duration, the viscous regime being quickly reached. Perhaps I missed something and the authors already mentioned this, but I could not find it.
3. I do not fully understand the notation ΣF_{ij} . I admit that the interactions can be reduced to pair interactions and that n-body interactions can be neglected, but the physical ingredients contained in this term are not clear. I assume that it contains electric/electrophoretic mutual interactions and also hydrodynamic interactions. Is that all? anyway, it should be specified.

Reviewer #2 (Remarks to the Author):

Review of NCOMMS-19-11248856A

This revised version of the manuscript presents a much stronger understanding of the underlying phenomena present in the system. The study now shows a number of simulations involving different forces and a wide range of models to represent a variety of particles behavior under the different conditions evaluated. The authors also addressed the reviewer's concerns regarding reproducibility of experiments (standard deviation and repetitions) and the large-area robustness of the device by evaluating particle (SUVs) capture over multiple trapping cell units using FL micrographs and statistical analysis. The manuscript now looks more organized and many details and additional information has been included in the supplementary material that allows for a better understanding of the different simulation models employed to explain what was observed experimentally.

What this reviewer considers that still need to be addressed is that the figure section labels ("a", "b", "c", ...) were often hard to follow when they were presented in white with the simulation images next to white arrows (i.e. Fig 1d,e,h,i, and Fig 2a-e) and where the label is obstructed or small enough to be mistaken with other information from the images.

POINT-BY-POINT RESPONSE TO REVIEWERS

MANUSCRIPT INFO

Manuscript ID: Nature Communications Manuscript NCOMMS-19-1124856

Title: "Precise capture and dynamic relocation of nanoparticulate biomolecules through dielectrophoretic enhancement by vertical nanogap architectures"

CONTENTS

Point-by-point response to comments from reviewer # 1: p. 1

Point-by-point response to comments from reviewer # 2: p. 5

Additional Changes: p. 6

FORMATTING KEY

Reviewers' comments — ***Bold Italic Font***

Author Response— Regular Font

Author Actions— *Italic*

Text Changed —Normal Underlined Font

1. POINT-BY-POINT RESPONSE TO COMMENTS FROM REVIEWER # 1

Comments: *“The revised version of the manuscript has been definitely improved and the objective pursued by the authors now appears more clearly to me. Considering that the article has possible interesting applications in microfluidics and bio/nanotechnologies, I reckon that the article could be suitable for publication in Nature Communication. However, I still have some minor comments that should be addressed.”*

1. *“The authors define a potential for a dissipative force or a set of dissipative forces (hydrodynamic interactions with the EO flow and DEP mainly). While I understand why they are doing this, they should mention that this is not a real potential (forces are not conservative). They could use the term ‘pseudo-potential’ for instance to stress that this potential is not defined in a rigorous way mechanically speaking.”*

[Author response] We appreciate the Reviewer’s suggestion for concern that a set of forces suggested in the manuscript are not conservative. As reviewer #1 suggested, a terminology of ‘pseudo-potential (which is also a dissipation function)’ is a more rigorous and accurate representation of our interpretations to characterize the role of dissipative (non-conservative) forces acting on localization of particles. Now in the revised manuscript, the author uses ‘pseudo-potential’ instead of ‘potential’, and its relevant explanation were newly addressed both in main manuscript in briefly and supplementary information with depth explanation.

[Author actions] *Explanation on introducing ‘pseudo-potential’ was newly included and words ‘potential’ were revised into ‘pseudo-potential’.*

TEXT ADDED/REVISED: Main manuscript Page 9

“Then, because the ACEO-induced slip velocity (pushing toward centre) is found to be stronger than the DEP force (attracting toward edges) at the collecting surface of the VNE, effective surface streams push particles toward the stagnation point where the **pseudo**-potential energy is the lowest (**Supplementary Note 12**)⁵¹. In contrast, at 100 kHz (**Fig. 3h**) and 1 MHz (**Fig. 3i**) in regime II, the ring-shaped particle assembly was observed at the edge of nanogap, where the value of ∇E^2 is the largest and the **pseudo**-potential energy is the lowest (**Supplementary Section 11**).”

TEXT ADDED/REVISED: Supplementary information Page 18

“**Supplementary Note 11: Pseudo-potential for localization of particles.**

Particles suspended in an aqueous environment move and migrate to positions at which the potential energy is minimum. **In this approach with non-conservative forces, the particle behaviour of pseudo-potential was introduced to interpret role of dissipative forces on localizing behaviours of particles.** As two dominant forces affect the local position of the suspended particles on the VNE surface, the total **pseudo**-potential energy profile along the surface, $U(x)$, can be given as $U(x) = U_{\text{DEP}}(x) + U_{\text{ACEO}}(x)$, where U_{DEP} and U_{ACEO} indicate that the **pseudo**-potential energies arise from DEP- and ACEO driving forces. Note that two dominant **pseudo**-potential energies are achieved from the inverse gradient of \mathbf{F}_{DEP} and \mathbf{F}_{ACEO} using the equations $U_{\text{DEP}} = -\pi\epsilon_m R^3 \text{Re}[f_{\text{CM}}(\omega)]\mathbf{E}^2$ and $\mathbf{F}_{\text{ACEO}} = -\nabla U_{\text{ACEO}}$, respectively^{23,28}. Shown in **Supplementary Fig. 10**, the minimum value of $U(x)$ is calculated at the centre of the VNE at $f = 10$ kHz (**Supplementary Fig. 10a**), in contrast to the edges of the VNE at $f = 100$ kHz (**Supplementary Fig. 10b**), respectively. This shows great consistency with the experimental demonstration shown in **Main Fig. 3**.”

FIGURE LEGEND ADDED/REVISED: Supplementary Figure 19

“**Supplementary Figure 10. Energy landscape of pseudo-potential on VNE surface.**
a, b Calculated $U_{\text{DEP}}(x)$ (red), $U_{\text{ACEO}}(x)$ (blue), and $U(x)$ (black) along the bottom electrode surface of VNE ($d = 100$ nm) under $V_{\text{pp}} = 2.5$ V, $f = 10$ kHz (**a**), and 100 kHz (**b**).”

2. “*The Langevin equation is written in the inertial full form. The authors should say early in the derivation (actually as soon as they write the equation) that they are aware that the ballistic regime is of short duration, the viscous regime being quickly reached. Perhaps I missed something and the authors already mentioned this, but I could not find it.*”

[**Author response**] We appreciate the Reviewer’s comment. As reviewer #1 pointed out, transient ballistic regime of Langevin equation (short time scale of $t \ll \tau$) reaches into saturated viscous regime (long time scale of $t \gg \tau$) where τ denote relaxation time of particles. Since the τ of the particle is calculated to be much smaller than the typical time of observation, ballistic regime quickly reaches into viscous regime. For time scale of viscous regime ($t \gg \tau$), $e^{-t/\tau}$ term can be negligible and the equation is saturated into right term in the equation (4). The authors newly added and revised comments for this issue in the main manuscript.

[Author actions] Comments on relaxation time and transition from ballistic regime into viscous regime were newly included in the main manuscript.

TEXT ADDED/REVISED: Main manuscript Page 5

“Since the relaxation time (τ) of the particle ($\tau = m/6\pi\eta R \approx 6 \times 10^{-8}$ s) is much smaller than the typical experimental observation time (t), transient ballistic regime of short time scales ($t \ll \tau$) quickly reaches viscous regime of a time scale longer than τ ($t \gg \tau$). Thus, equation (2) can be approximated into terminal \mathbf{u}_p of viscous regime as³⁶

$$\mathbf{u}_p = \left(\mathbf{u}_{\text{ACEO}} + \frac{\mathbf{F}_{\text{DEP}}}{6\pi\eta R} \right) (1 - e^{-t/\tau}) \approx \mathbf{u}_{\text{ACEO}} + \frac{\mathbf{F}_{\text{DEP}}}{6\pi\eta R}. \quad (1)$$

with the assumption that the initial velocity of the particle is 0.”

3. “I do not fully understand the notation ΣF_{ij} . I admit that the interactions can be reduced to pair interactions and that n -body interactions can be neglected, but the physical ingredients contained in this term are not clear. I assume that it contains electric/electrophoretic mutual interactions and also hydrodynamic interactions. Is that all? anyway, it should be specified.”

[Author response] The authors thank Reviewer #1 for the comments to improve our work adding a depth of rigorous analysis. We now agree with Reviewer’s opinion that notating particle interaction in a previously revised form of ‘ ΣF_{ij} ’ can cause confusions to the readers since the Langevin equation has been defined by forces acting on a single particle. In revised version, we replace the ‘ ΣF_{ij} ’ to ‘ \mathbf{F}_{int} ’, which is described as a Columb interparticle interaction force to define particle-particle interaction acting on a single particle. This notation follows the reference, Lapitsky, D. S. Particle separation by alternating electric fields of quadrupole type. *J. Phys.: Conf. Ser.* **774**, 012178 (2016).

[Author actions] The notation was revised in to ‘ \mathbf{F}_{int} ’ and the following notations are also revised accordingly.

TEXT ADDED/REVISED: Main manuscript Page 4-5

“To evaluate the competitive forces and resultant dynamics of a spherical particle (with its mass of m_p) in an aqueous environment, the Langevin equation of particle velocity (\mathbf{u}_p) was employed³⁵:

$$m_p \frac{d\mathbf{u}_p}{dt} = \mathbf{F}_{\text{DEP}} + \mathbf{F}_{\text{ACEO}} + \mathbf{F}_{\text{ETF}} + \mathbf{F}_{\text{grav}} + \mathbf{F}_{\text{buoy}} + \mathbf{F}_{\text{int}} + \xi(t) \quad (2)$$

considering DEP force (\mathbf{F}_{DEP}), Stokes drag force induced by ACEO flow (\mathbf{F}_{ACEO}) and electrothermal flow (\mathbf{F}_{ETF}), gravitational force (\mathbf{F}_{grav}), buoyant force (\mathbf{F}_{buoy}), interparticle force from Coulomb interaction (\mathbf{F}_{int}), and random Brownian force ($\xi(t)$). Because Stokes drag forces are

defined as $\mathbf{F} = -6\pi\eta R(\mathbf{u}_p - \mathbf{u}_m)$, where \mathbf{u}_p and \mathbf{u}_m are the velocities of particles and fluidic flows, respectively, \mathbf{F}_{ACEO} and \mathbf{F}_{ETF} exerting on a single particle can be evaluated by inserting \mathbf{u}_{ACEO} and \mathbf{u}_{ETF} into \mathbf{u}_m , respectively³⁶. Considering the simulations (\mathbf{F}_{ETF} , \mathbf{F}_{DEP} , and \mathbf{F}_{ACEO}), calculations (\mathbf{F}_{grav} , \mathbf{F}_{buoy} , and $\zeta(t)$), and low concentration (10 ppm) condition (\mathbf{F}_{int}) on the VNE having pattern size (L) and periodicity (P) of $L = 10 \mu\text{m}$ and $P = 30 \mu\text{m}$ (**Fig. 1b**), \mathbf{F}_{DEP} and \mathbf{F}_{ACEO} dominantly determine the movement of 1- μm -diameter polystyrene (PS) particles, while the others are negligible (**Supplementary Notes 1-5**).”

Main manuscript Page 9

“Note that the interparticle force component (\mathbf{F}_{int}) is negligible when suspended particles are distant from each other.”

2. POINT-BY-POINT RESPONSE TO COMMENTS FROM REVIEWER #2

Comments: *“This revised version of the manuscript presents a much stronger understanding of the underlying phenomena present in the system. The study now shows a number of simulations involving different forces and a wide range of models to represent a variety of particles behavior under the different conditions evaluated. The authors also addressed the reviewer’s concerns regarding reproducibility of experiments (standard deviation and repetitions) and the large-area robustness of the device by evaluating particle (SUVs) capture over multiple trapping cell units using FL micrographs and statistical analysis. The manuscript now looks more organized and many details and additional information has been included in the supplementary material that allows for a better understanding of the different simulation models employed to explain what was observed experimentally.”*

1. *“What this reviewer considers that still need to be addressed is that the figure section labels (“a”, “b”, “c”, ...) were often hard to follow when they were presented in white with the simulation images next to white arrows (i.e. Fig 1d,e,h,i, and Fig 2a-e) and where the label is obstructed or small enough to be mistaken with other information from the images.”*

[Author response] The authors appreciate with reviewer #2 for providing constructive comments. As reviewer #2 pointed out, several figure labels are often hard to follow due to the poor visibility. Now in the revised version, the authors rearranged several labels in figures to improve the legibility of the manuscript. Not only figures pointed out by the reviewer #2, but also the other figures were also modified.

[Author actions] *Several labels were moved to elsewhere or borders are improved for better visibility.*

FIGURE REVISED:

Labels moved: Main Figs. 1d, 1e, 1h, 1i, 5f-i / Supplementary Figs 5a, 5b, 6a, 6b, 8, 11a-d

Borders improved: Main Figs. 2c, 3a-e, 4e, 5a, 5e / Supplementary Figs 9, 11a-d, 12a, 13b, 13c, 16a, 20a, 20b, 22c